# Exploiting Latent Properties to Optimize Neural Codecs

## Abstract

End-to-end image/video codecs are getting competitive compared to traditional compression techniques that have been developed through decades of manual engineering efforts. These trainable codecs have many advantages over traditional techniques such as an easy adaptation on perceptual distortion metrics and high performance on specific domains thanks to their learning ability. However, current state-of-the-art neural codecs do not fully exploit the benefits of vector quantization and the existence of the gradient of entropy in the decoding device. In this research, we propose leveraging these two properties to improve the performance of off-the-shelf codecs. Firstly, we demonstrate that using non-uniform scalar quantization cannot improve performance over uniform quantization. Thus, we suggest using a predefined optimal uniform vector quantization to improve performance. Secondly, we show that the gradient of entropy available at the decoder side is correlated with the gradient of the reconstruction error, which is not available at the decoder side. Thus, we utilize the former as a proxy to enhance compression performance. Our experimental results show that these approaches can save between 2-4% of the rate for the same quality across various pre-trained methods.

## 1 Introduction

Lossy Image and video compression is a fundamental task in image processing which has become crucial in the time of the pandemic and the increasing volume of video streaming. Thanks to the community's decades long efforts, traditional methods (e.g. VVC) have reached current state of the art rate-distortion (RD) performance and dominate the current codecs market. Recently, end-to-end trainable deep models have emerged with promising RD performances by learning the informative latents and modeling the latent distribution. Even though deep learning based models clearly exceed many traditional techniques and surpass human capability for some general computer vision tasks, they are only slightly better than the best traditional codecs on single image compression according to our knowledge.

End-to-end deep compression methods typically refer to rate-distortion autoencoders (Habibian et al., 2019), in which the latents are generated by jointly optimizing the encoder, decoder, and entropy model with a rate-distortion loss function. For perceptual friendly compression, distortion based on a perceptual metric can also be used in the loss function (Blau & Michaeli, 2019). These methods can be seen as a special case of Variational Autoencoder (VAE) models as described in (Kingma & Welling, 2013), where the approximate posterior distribution is a uniform distribution centered on the encoder's outputs (latents) at training time and has a fixed variance output distribution and trainable priors (Theis et al., 2017; Ballé et al., 2017). It was shown that minimizing the evidence lower bound (ELBO) of this special VAE is equivalent to jointly minimizing the mean square error (MSE) of the reconstruction and the entropy of the latents w.r.t the priors (Ballé et al., 2018). All of the proposed models mainly differ in the way they model priors: using either fully-factorized (Ballé et al., 2017), zero-mean Gaussian (Ballé et al., 2018), Gaussian (Minnen et al., 2018; Minnen & Singh, 2020) or a mixture of Gaussian (Cheng et al., 2020), where some methods predict the priors using an autoregressive schema (Minnen et al., 2018; Minnen & Singh, 2020; Cheng et al., 2020; Xie et al., 2021; He et al., 2021) and some improve the priors through global and local context modeling (Qian et al., 2021; Kim et al., 2022). These neural image codecs were extended to the video compression domain by using two VAEs, one for encoding motion information and another one for encoding residual information

in end-to-end video compression (Lu et al., 2019; Agustsson et al., 2020; Ladune & Philippe, 2022; Yılmaz & Tekalp, 2021; Pourreza & Cohen, 2021; Li et al., 2021; 2022)

An important step in building a neural codec is the quantization of the latents before the entropy coding. Nearly all of the mentioned prior state of the art models use a fixed bin-width uniform Scalar Quantization (SQ). Although Vector Quantization (VQ) is theoretically better (Gersho & Gray, 2012), there have been very few attempts to use VQ in neural codecs such as in (Agustsson et al., 2017; Zhu et al., 2022). However, these attempts did not show improvement over SQ, which can be attributed to several reasons. Firstly, VQ introduces extra trainable parameters for learning codeword centroids. Learning these parameters jointly with all model parameters can introduce additional complexity at training time. Secondly, since quantization is non-differentiable, a relaxation strategy (Bengio et al., 2013; Agustsson et al., 2017; Zhu et al., 2022) is needed during training time, and the continuous relaxation of VQ is less efficient than that of SQ.

One of the main principle in general compression is to exploit all available information at the decoder to reconstruct the data. Surprisingly, even though the gradients of the entropy w.r.t latents are available at the decoder side, this information remains unused so far in the literature. Some similar works in the literature attempted to improve the performance of the codec during encoding, for example by using specific parameterization (Balcilar et al., 2022), or computationally heavy finetuning solutions (Yang et al., 2020; Guo et al., 2021), either partially (Campos et al., 2019; Lu et al., 2020) or entirely (van Rozendaal et al., 2021). However, all of these methods disregard the existence of the gradient of the entropy in decoding.

In this paper, we present two novel contributions aimed at leveraging the properties of learned latent representations for compression. Firstly, we demonstrate that any kind of non-uniform scalar quantization cannot improve performance compare to uniform scalar quantization, if the neural model is enough expressive. Thus, we eliminate scalar quantization and propose to use uniform vector quantization over the latents. Since the optimal uniform VQ map is known up to some certain dimensions, learning partitioning is not needed. This contribution can be applied even without re-training the model if the original model is trained for uniform SQ, which is often the case for neural codecs.

Secondly, we apply the Karush–Kuhn–Tucker (KKT) conditions on the neural codec which has not been done to the best of our knowledge. These conditions reveal a connection between the gradient of the reconstruction error (unavailable at the decoder side) and the gradient of the entropy (available at the decoder side). This finding motivated us to test correlations between these two gradients. Since we find strong correlation between two gradients for various neural codecs, we use the available one as a proxy for the unavailable one to improve the performance of neural codecs without requiring re-training. Our two contributions are generic enough (not depending on the encoder-decoder architecture) to achieve a rate saving of 2-4% at the same quality for several neural codec architectures.

## 2 Problem statement and State of the Art

As shown in Figure 1, given an input color image $\mathbf{x} \in \mathbb{R}^{n \times n \times 3}$ to be compressed (the image can be considered square without any loss of generality), the neural codec learns a non-linear encoder $g_a(\mathbf{x}; \phi)$, parameterized by the weights $\phi$. The output of the encoder, $\mathbf{y} \in \mathbb{R}^{m \times m \times o}$, is called the main embeddings (or main latents) of the image. The latent representation is then quantized as $\tilde{\mathbf{y}} = Q(\mathbf{y})$ to obtain the main codes of the image. The dequantization block $\hat{\mathbf{y}} = Q^{-1}(\tilde{\mathbf{y}})$ is used to obtain reconstructed main latents $\hat{\mathbf{y}}$ at the decoding side [1]. The decompressed image $\hat{\mathbf{x}} \in \mathbb{R}^{n \times n \times 3}$ is obtained by the learned deep decoder with $\hat{\mathbf{x}} = g_s(\hat{\mathbf{y}}; \theta)$. The neural codec is learned to minimize two objectives simultaneously, namely the distortion between $\mathbf{x}$ and $\hat{\mathbf{x}}$, and the length of the bitstream needed to encode $\tilde{\mathbf{y}}$. The codes are losslessly encoded into a bitstream using an entropy encoder such as range asymmetric numeral systems (RANS) Duda (2009). RANS requires the probability mass function (PMF) of each code, and which is also learned at training time. Since RANS is asymptotically optimal, the lower bound of bitlength according to Shanon's entropy theorem can be used instead of experimental bitlength from RANS in order to make the bitlength objective differentiable. Thus,

---

[1]The dequantization stage is sometimes skipped in practice since it can be done inside the first linear operation of the decoder.

neural codecs use the entropy model to learn the PMF of each codes under determined quantization, which allows us to determine the lower bound of the bitlength.

Current state of the art neural codecs use a hyperprior entropy model, where the side embedding (or side latents) $\mathbf{z} \in \mathbb{R}^{t \times t \times s}$ is learned by another deep neural network by $\mathbf{z} = h_a(\mathbf{y}; \Phi)$. The side embeddings are quantized as $\tilde{\mathbf{z}} = Q(\mathbf{z})$ to obtain side codes $\tilde{\mathbf{z}}$ followed by dequantization $\hat{\mathbf{z}} = Q^{-1}(\tilde{\mathbf{z}})$ to obtain the reconstructed side latents $\hat{\mathbf{z}}$. The main motivation of side information is to remove any image structure that would be left in the main latent representation $y$. The hyperprior entropy model assumes that the probability density function (PDF) of each scalar latent follows a Gaussian distribution, where the parameters are obtained by another deep network such as $\boldsymbol{\mu}, \boldsymbol{\sigma} = h_s(\hat{\mathbf{z}}; \Theta))$ [2]. Thus, the hyperprior model's prediction can be defined by $p_{\hat{\mathbf{y}}_{ijk}}(.) := \mathcal{N}(.|\boldsymbol{\mu}_{ijk}, \boldsymbol{\sigma}_{ijk})$. The PMF of latent code $P(\tilde{\mathbf{y}}_{ijk})$ can be written under determined quantization as a function of $p_{\hat{\mathbf{y}}_{ijk}}(.)$. Using this PMFs, the lower bound of the main codes' bitlength can be defined by $-\log(p_h(\hat{\mathbf{y}}; \hat{\mathbf{z}}, \Theta)) := -\sum_{ijk} \log(P(\tilde{\mathbf{y}}_{ijk}))$. The factorized entropy model learns the PDF for each $t \times t$ slice of latent defined as $p_{\hat{\mathbf{z}}_{:,:,s}}(.)$. This PDF is enough to have the PMF of side codes $P(\hat{\mathbf{z}}_{ijk})$ under determined quantization. Thus, the lower bound of side codes' bitlength can be defined by $-\log(p_f(\hat{\mathbf{z}}; \Psi)) := -\sum_{ijk} \log(P(\tilde{\mathbf{z}}_{ijk}))$.

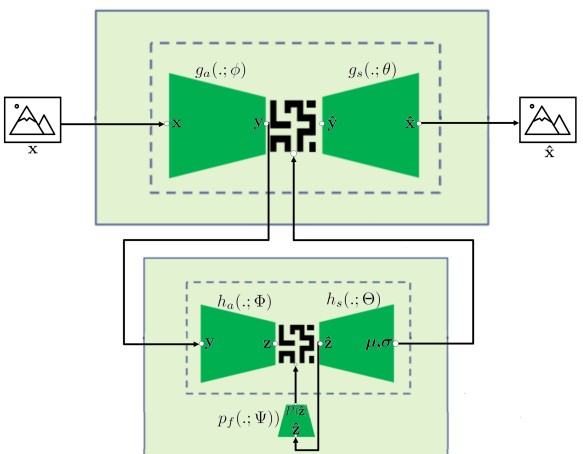

Figure 1: Block diagram of the state-of-the art neural codecs. Five dark green blocks are the trainable blocks implemented by neural networks, while binary patterns show the quantization and entropy encoding/decoding process driven by certain entropy model's PMFs on main and side latents.

In this setting, the deep encoder ($g_a(.; \phi)$), the deep decoder ($g_s(.; \theta)$), the hyperprior entropy model $p_h(.; \hat{\mathbf{z}}, \Theta)$ (composed of the deep hyperprior encoder ($h_a(.; \Phi)$), the deep hyperprior decoder ($h_s(.; \Theta)$), and the factorized entropy model $p_f(.; \Psi)$) are the trainable blocks implemented by neural networks. Each block with its trainable parameter, input and output are depicted in Figure 1. The optimal values of the parameters $\phi, \theta, \Phi, \Theta$ and $\Psi$ are found by minimizing the following loss function using the training sample $\mathbf{x}$.

$$\mathcal{L} = \mathop{\mathbb{E}}_{\mathbf{x} \sim p_x} \left[ -\log(p_f(\hat{\mathbf{z}}; \Psi)) - \log(p_h(\hat{\mathbf{y}}; \hat{\mathbf{z}}, \Theta)) + \lambda d(\mathbf{x}, \hat{\mathbf{x}}) \right], \tag{1}$$

where $d(.,.)$ is a distortion measure between the original and the reconstructed image (for example the mean square error). The rate term is the sum of the lower bound of bitlength of the side information ($-\log(p_f(\hat{\mathbf{z}}; \Psi))$) and the main information ($-\log(p_h(\hat{\mathbf{y}}; \hat{\mathbf{z}}, \Theta))$). Hyperparameter $\lambda$ controls the trade-off between the rate (r) and distortion (d) terms.

Both the quantization step $Q(.)$ and its counterpart dequantization $Q^{-1}(.)$ need to be applied for the main and side information. In addition, both the factorized and hyperprior entropy models should know the determined quantization technique to have the PMF. To the best of our knowledge, most of the current methods implement quantization as a 1-bin width uniform Scalar Quantization (SQ), with few exceptions (Agustsson et al., 2017; Agustsson & Theis, 2020; Guo et al., 2021). This quantization step is implemented by element-wise nearest integer rounding $Q(x) = round(x)$ and its dequantization $Q^{-1}(x) = x$. Thus, the PMF of $\tilde{\mathbf{y}}_{ijk} \in \mathbb{R}$ can be calculated by $P(\tilde{\mathbf{y}}_{ijk}) = \int_{\hat{\mathbf{y}}_{ijk}-0.5}^{\hat{\mathbf{y}}_{ijk}+0.5} p_{\hat{\mathbf{y}}_{ijk}}(x)dx$ where $p_{\hat{\mathbf{y}}_{ijk}}(.)$ is the PDF of latent $\hat{\mathbf{y}}_{ijk}$ learnt by entropy model [3]. Since the nearest integer rounding operation has non-informative gradients,

---

[2]In autoregressive prediction, they are predicted step by step using the previous main latents in a sense of being previous part of the local neighborhood by $\boldsymbol{\mu_i}, \boldsymbol{\sigma_i} = h_s(\hat{\mathbf{z}}, \hat{\mathbf{y}}_{<\mathbf{i}}; \Theta)$

[3]Entropy model can alternatively learn cumulative distribution function CDF, $\sigma_{\hat{\mathbf{y}}_{ijk}}(x)$ instead of PDF and calculate PMF by $P(\tilde{\mathbf{y}}_{ijk}) = \sigma_{\hat{\mathbf{y}}_{ijk}}(\hat{\mathbf{y}}_{ijk} + 0.5) - \sigma_{\hat{\mathbf{y}}_{ijk}}(\hat{\mathbf{y}}_{ijk} - 0.5)$

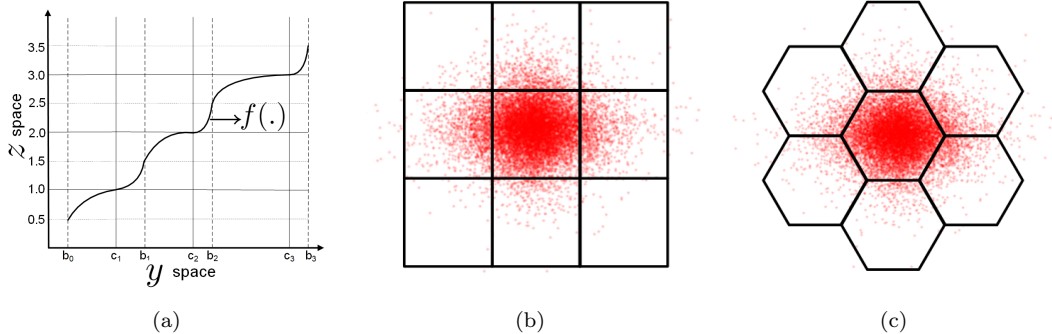

(a)  (b)  (c)

Figure 2: a) $f : y \to z$ transforms non-uniform quantization map (grid borders $b_0, \ldots b_n$ and grid centers $c_1, \ldots c_n$) to uniform map (centers are located on integer where borders are at the middle of two consecutive centers). b) Uniform SQ grids on 2D c) Optimal uniform VQ grids on 2D.

a continuous relaxation must be applied at training as $Q(x) = x + \epsilon$, where $\epsilon$ is randomly sampled from the uniform distribution $\epsilon \sim U(-0.5, 0.5)$.

However, for the Vector Quantization (VQ) case, latents can be packed into a $v$-dimensional vector $u \in \mathbb{R}^v$ and each $u$ is assigned to a single code. The quantization centers $C_j \in \mathbb{R}^v, j = 1 \ldots M$ are learned by the entropy model. Thus, the quantization step finds the index of the nearest center as $Q(x) = \arg\min_i \|x - C_i\|$, while dequantization returns the quanta center as $Q^{-1}(i) = C_i$ where $i \in \{1 \ldots M\}$. In this case, there are $M$ different quantization centers, and $M$ unique codes. Since the argmin operator applies hard assignment, it has non-informative gradients and continuous relaxation must also be applied during training. This is generally achieved by softmax operator that assigns all codes to the latent vector with different probabilities, depending on the distances to the centers. These probabilities are used by the entropy model to learn the PMF of each quanta center (codes), and also for dequantization which is the expectation of quanta centers under these probabilities during training.

## 3 Uniform Vector Quantization

Uniform SQ which is widely used in neural codecs is not the optimal quantizer among all SQs. In fact, it is known that the optimal quantizer should have smaller grid sizes in regions of higher probability and larger grid sizes in regions of lower probability (Farvardin & Modestino, 1984). Thus, a non-uniform SQ which is aware of source distributions can have lower quantization error than uniform SQ. However, we state in the following theorem that uniform SQ is sufficient among all SQs in the neural codecs.

**Theorem 1.** *If a neural codec has an encoder block $g_a : \mathbb{R}^{n \times n \times 3} \to \mathbb{R}^{m \times m \times o}$, an decoder block $g_s : \mathbb{R}^{m \times m \times o} \to \mathbb{R}^{n \times n \times 3}$ and it needs a non-uniform SQ map for the optimal rate-distortion performance, there exists another neural codec that gives the same rate-distortion performance with 1-bin width uniform SQ (nearest integer rounding quantization) whose encoder block is $f \circ g_a$ and decoder block is $g_s \circ f^{-1}$ where $f : \mathbb{R}^{m \times m \times o} \to \mathbb{R}^{m \times m \times o}$ is an invertible transformation.*

The proof can be found in Appendix A and is based on modelling the one-dimensional quantizer using a memoryless monotonically increasing nonlinearity followed by a uniform fixed-point quantizer as in (Bennett, 1948). We show that an invertible function that can be implemented by neural networks can transform the borders and grid centers of a non-uniform quantization map to a uniform quantization map. A simple illustration of this is shown in Figure 2(a). According to Theorem 1 neural codecs with nearest integer rounding quantization is sufficient among all scalar quantizations as long as it is enough expressive, that its encoder has invertible layers at the bottom of the encoder block and also the inverse of this layers should be at the beginning of the decoder block. Most of the neural codecs do not have invertible blocks, but since the decoder is almost the inverse of the encoder (in a sense that it is able to revert back image with negligible distortion) and usually they are deep enough with millions of parameters, we can assume that they are

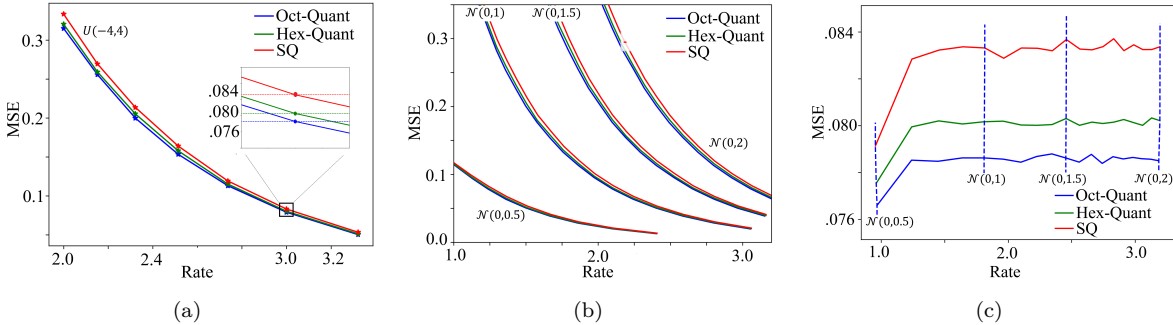

Figure 3: RD performances of different volume uniform SQ, regular hexagon **Hex-Quant** and truncated octahedron **Oct-Quant** grid. a) Uniform source is sampled from $U(-4, 4)$ and zoom-in where the grid volume is unitary. b) Different Gaussian sources. c) RD plot of unitary volume grids for Gaussian sources.

enough expressive. The advantage of using invertible layers between less powerful encoder and decoder block was experimentally found recently in (Shukor et al., 2022). Also, the sufficiency of uniform quantization in neural codecs was discussed in (Ballé et al., 2021). Our theorem verifies these two prior contributions.

By the result of Theorem 1, using non-uniform SQ in neural codec is meaningless. One straight-forward selection for lower quantization error is VQ. Even though VQ is theoretically optimal than SQ even if the dimensions are i.i.d (Gersho, 1982), so far it could not perform significantly better than uniform SQ in neural codecs. The reasons could be that (i) VQ learns non-uniform partitioning where the convergence of these centers might encounter training difficulties and can take longer training time whereas in uniform quantization, the quantization centers are predefined and fixed and (ii) since the dequantization of VQ averages these variable quantization centers during training phase, the reconstructed latents in training phase may be far away from their hard assignment values (in test phase). This mismatch can be higher than mismatch between nearest integer rounding and its continuous relaxation by additive uniform noise. On the other hand, uniform VQ can solve above mentioned problems by using predefined uniform partitioning.

### 3.1 Space tessellation grids

In this research, we propose to use uniform vector quantization that is based on space tessellation. This method involves using a predefined high dimensional grid to cover the entire high-dimensional space. Thus, the quantization grids are fixed instead of learning it. The nearest integer quantization grid in 1D is equivalent to a unit square grid in 2D, as shown in Figure 2(b) for a zero-mean Gaussian source, however it might not be the best partition of the space with uniform grids, thus using nearest integer quantization (i.e square grids in 2D) directly in the neural codec might not yield better RD performance. Following remark shows the existence of entropy constrained optimal space tesellation and can be used for uniform vector quantization.

**Remark 1.** *VQ constraints with a uniform grid produces the optimal space tessellation which refers to a pattern of v-dimensional shapes that fit perfectly together without any gaps and have minimum inertia. The optimal shapes are regular hexagon for 2D and truncated octohedron for 3D case. (Gersho, 1979).*

The above remark shows that instead of learning the grids centre and optimal partition, we can use the optimal space tessellation grids at the corresponding dimensions. In this paper our proposal for 2D space is to use a regular hexagon grid that has unitary volume as shown in Figure 2(c). Theoretical advantage of space tessellation grids compared to the nearest integer quantization is shown in Appendix B for a uniform distribution. We also showcase the simulation results under uniform and Gaussian sources in Figure 3a-b. For these simulations, we generate random numbers under a uniform distribution $U(-4, 4)$ for Figure 3a, and four different zero-mean Gaussian distributions with specified scales for Figure 3b. We quantize these numbers using three different methods: uniform SQ (**SQ**), regular hexagonal grid (**Hex-Quant**), and truncated octahedron grid (**Oct-Quant**), with varying grid sizes. The left side of the $x$-axis represents higher rates where the grid size is small, while the right side represents smaller rates where the grid size is big. In Figure 3a,

we zoom in on the rate where the grid size is unitary. The MSE of the three different quantization grids is very close to their theoretically known MSE, which are 0.0833, 0.0801, and 0.0785, as shown in Appendix B. For Figure 3c, we generate random numbers from a zero-mean Gaussian distribution with different scales in the range of $[0.5, 2]$, with a 0.1 interval between scales. We quantize each source using a unitary volume grid of the three quantization maps mentioned above. The left side of the x-axis represents low rates when a source from smaller scale Gaussian distributions is used, while the right side represents higher rates when a source generated by larger scale Gaussian distributions is used. For all three methods, the same source does not necessarily have the same rate under a unitary grid size. Thus, the vertical dashed lines representing the four different distributions are not aligned well. This simulation is particularly important, because the main information in modern neural codec is zero mean Gaussian distribution with different scale and is quantized with unitary grids. Our simulation demonstrates that, for the same bitrate, the uniform vector quantization (VQ) grid has lower mean squared error (MSE) than SQ. This simulation emphasizes the importance of using uniform VQ as a more effective method for quantization in neural codecs. Now we show that how our uniform vector quantization can be applied in the off-the-shelf neural codec without re-training.

## 3.2 Uniform VQ with off-the-self neural codec

Let $\mathbf{y} \in \mathbb{R}^{m \times m \times o}$ be the latents of the deep encoder, $v$ be the dimension where the optimal space tessellation is performed, $\mathbf{c}^{(i)} \in \mathbb{R}^v, i = 1 \ldots M$ be the $M$ grid centers of the $v$-dimensional shape and $\mathbf{c}_j^{(i)} \in \mathbb{R}$ is the scalar value on $j$-th dimension of the center $\mathbf{c}^{(i)}$. In order to quantize the latents, we reshape the latents $\mathbf{y}$ into pseudo $v$-dimensional latents such that $\mathbf{y} \in \mathbb{R}^{m \times m \times o} \to \check{\mathbf{y}} \in \mathbb{R}^{b \times v}$, where $b = \frac{m.m.o}{v}$. The quantization is performed by assigning the nearest neighbours grid centers, as $\tilde{\mathbf{y}}_j = Q(\check{\mathbf{y}}_j) = \arg\min_i ||\check{\mathbf{y}}_j - \mathbf{c}^{(i)}||$, where $\tilde{\mathbf{y}}_j \in \{1 \ldots M\}, j = 1 \ldots b$ are the codes to be encoded into a bitstream. In practice, instead of reshaping all latents together into $v$ dimension, we can reshape the latents whose learned PDFs are the same (i.e coming from the same distribution). This approach does not have significantly different results, only needs smaller PMF table to be kept in decoding device.

To encode the codes ($\tilde{\mathbf{y}}$) into a bit-stream, we cannot use the off-the-shelf neural codec's 1D entropy model's PMF calculation, as our latents are $v$-dimensional latents and domain is different. To this end, we propose to create the PMF by integrating the pseudo-$v$ dimensional PDF ($\prod_{j=1}^{v} p_{\mathbf{c}_j^{(i)}}(u_j)$) inside the grid $G$ whose center is located on $\mathbf{c}^{(i)}$:

$$P(\mathbf{c}^{(i)}) = \int_{G+\mathbf{c}^{(i)}} \left( \prod_{j=1}^{v} p_{\mathbf{c}_j^{(i)}}(u_j) \right) du \tag{2}$$

Here $P(\mathbf{c}^{(i)})$ is the PMF of $i$-th symbol in the dictionary and $p_{\mathbf{c}_j^{(i)}} : \mathbb{R} \to \mathbb{R}$ is PDF of corresponding latent, learned by entropy model in the baseline model. Since there is no closed form solution for the integral of multi dimensional Gaussian distribution over a hexagonal and truncated octahedron domain (Savaux & Le Magoarou, 2020), we use numeric solvers to approximate the solution of equation 2. We can reach numeric solution for main codes PMFs where $p_{\mathbf{c}_j^{(i)}}(.)$ is the Gaussian distribution and also non-parametric $p_{\mathbf{c}_j^{(i)}}(.)$ for side code's PMF. In Appendix E, we show how to calculate equation 2 for the hexagonal domain. Another way to calculate those integral is by monte-carlo simulation for known PDFs $p_{\mathbf{c}_j^{(i)}}(.)$ before deployment, which we use for the truncated octahedron domain.

## 4 Forgotten Information: Gradient of Entropy

Receiver can compute the gradients of the main/side entropy w.r.t the main/side latents after decoding the main/side latents. Since these gradients are only available after decoding the latent codes, they may not seem useful at the first sight. This could be the reason why so far these gradients are never used in the testing phase. Here, we propose to use these gradients through the analysis of the Karush-Kuhn-Tucker conditions, and we experimentally find that they can be correlated with other useful gradients.

We can see the neural codec's loss function in equation 1 as an unconstrained multi-objective optimization problem. The optimal solution of such a problem is called Pareto Optimal which is a solution where no objective can be made better off without making at least one objective worse off (Miettinen, 2012). The following remark shows a useful property of a solution of unconstrained multi-objective optimization.

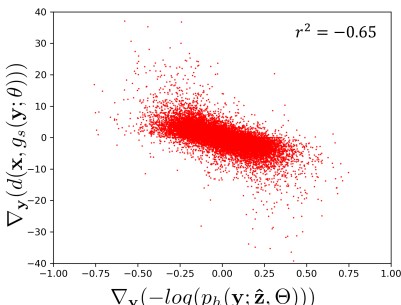

**Remark 2.** ***Karush–Kuhn–Tucker (KKT) conditions:*** *If the problem is $w^* = \arg\min_w(\sum_i \alpha_i . \mathcal{L}_i(w))$; where $\alpha_i \geq 0$, $\sum \alpha_i = 1$ and $\mathcal{L}_i$ is the i-th objective to be minimized, the solution $w^*$ is pareto optimal if and only if it satisfies $\sum_i \alpha_i \nabla_w \mathcal{L}_i(w^*)) = 0$. (Désidéri, 2012).*

This remark is also valid for end-to-end compression models. Following theorem shows how to leverage KKT conditions for the end-to-end image compression models.

Figure 4: Correlation between gradients of the entropy and the reconstruction error w.r.t the main latents.

**Theorem 2.** *If an end-to-end compression model is Pareto Optimal, it satisfies following conditions.*

$$\mathbb{E}_{\mathbf{x} \sim p_x} \left[ \nabla_\Phi \log(p_f(\hat{\mathbf{z}}; \Psi)) + \nabla_\Phi \log(p_h(\hat{\mathbf{y}}; \hat{\mathbf{z}}, \Theta)) \right] = 0. \tag{3}$$

$$\mathbb{E}_{\mathbf{x} \sim p_x} \left( \nabla_\phi \left[ -\log(p_f(\hat{\mathbf{z}}; \Psi)) - \log(p_h(\hat{\mathbf{y}}; \hat{\mathbf{z}}, \Theta)) \right] + \lambda \nabla_\phi d(\mathbf{x}, g_s(\hat{\mathbf{y}}; \theta)) \right) = 0 \tag{4}$$

The proof is in Appendix C.1. Following corollary shows it's usefulness under a certain assumption.

**Corollary 2.1.** *If we assume that equation 3 and 4 are valid for every single input $\mathbf{x}$ independently, then an optimal end-to-end compression model satisfies the following conditions.*

$$\nabla_{\hat{\mathbf{z}}} \log(p_f(\hat{\mathbf{z}}; \Psi)) = -\nabla_{\hat{\mathbf{z}}} \log(p_h(\hat{\mathbf{y}}; \hat{\mathbf{z}}, \Theta)) \tag{5}$$

$$\nabla_{\hat{\mathbf{y}}} \left[ -\log(p_f(\hat{\mathbf{z}}; \Psi)) - \log(p_h(\hat{\mathbf{y}}; \hat{\mathbf{z}}, \Theta)) \right] = -\lambda \nabla_{\hat{\mathbf{y}}} d(\mathbf{x}, g_s(\hat{\mathbf{y}}; \theta)) \tag{6}$$

The proof can be found in Appendix C.2. Under this assumption, it implies that the gradient of the main and the side information's entropy w.r.t side latent show opposite directions in equation 5. Additionally, the gradient of the weighted reconstruction error and the gradient of the total entropy w.r.t main latent also show opposite directions in equation 6. Hence, under this assumption, we can claim that these pair of gradients have correlation coefficient of $-1$ and one can be used instead of another by simply changing the direction.

The presence of correlation has been validated for different neural codecs using different test sets, as presented in Appendix D and section 5. The scatter plot in Figure 4 illustrates the correlation between the gradient of the main entropy and the gradient of the reconstruction error w.r.t main latents for a specific image sample where the gradients were calculated on a neural codec described in Minnen et al. (2018). Based on our tests on various widely-used neural codecs, we have observed relatively strong correlations between the gradients in equation 6[4] (ranging from -0.1 to -0.5) and weaker correlations (ranging from -0.15 to 0.1) for the gradients in equation 5. Although, the discovered correlations are far from $-1$, they still hold significance. This discrepancy can be attributed to the fact that the assumption does not hold strongly.

**Latent Shift wrt Gradients.** By definition of gradient based optimization, $\hat{\mathbf{z}}$ needs to take a step in negative direction of $\nabla_{\hat{\mathbf{z}}}(-\log(p_h(\hat{\mathbf{y}}; \hat{\mathbf{z}}, \Theta)))$ in order to decrease main information bitlength $-\log(p_h(\hat{\mathbf{y}}; \hat{\mathbf{z}}, \Theta))$. However $\nabla_{\hat{\mathbf{z}}}(-\log(p_h(\hat{\mathbf{y}}; \hat{\mathbf{z}}, \Theta)))$ is not available before decoding $\hat{\mathbf{y}}$ in the decoding device, but at least the correlated gradient $\nabla_{\hat{\mathbf{z}}}(-\log(p_f(\hat{\mathbf{z}}; \Psi)))$ is known after decoding $\hat{\mathbf{z}}$. We claim that there is a real number step size $\rho_f^*$ that decrease the bitlength of main information such that;

$$-\log(p_h(\hat{\mathbf{y}}; \hat{\mathbf{z}}, \Theta) \geq -\log(p_h(\hat{\mathbf{y}}; \hat{\mathbf{z}} + \rho_f^* \nabla_{\hat{\mathbf{z}}}(-\log(p_f(\hat{\mathbf{z}}; \Psi)))), \Theta).$$

$\rho_f^*$ can be obtained through a search among candidates or by applying an optimization method such that;

$$\rho_f^* = \arg\min_{\rho_f} (-\log(p_h(\hat{\mathbf{y}}; \hat{\mathbf{z}} + \rho_f \nabla_{\hat{\mathbf{z}}}(-\log(p_f(\hat{\mathbf{z}}; \Psi)))), \Theta)). \tag{7}$$

---

[4]Since we find equivalent correlation, we use it as $\nabla_{\hat{\mathbf{y}}} \left[ -\log(p_h(\hat{\mathbf{y}}; \hat{\mathbf{z}}, \Theta)) \right] = -\lambda \nabla_{\hat{\mathbf{y}}} d(\mathbf{x}, g_s(\hat{\mathbf{y}}; \theta))$

For the second condition, $\hat{\mathbf{y}}$ needs to take a step in negative direction of $\nabla_{\hat{\mathbf{y}}}(d(\mathbf{x}, g_s(\hat{\mathbf{y}}; \theta)))$ in order to decrease reconstruction error $d(\mathbf{x}, g_s(\hat{\mathbf{y}}; \theta))$. $\nabla_{\hat{\mathbf{y}}}(d(\mathbf{x}, g_s(\hat{\mathbf{y}}; \theta)))$ is never available in the decoding device, but at least the correlated gradient $\nabla_{\hat{\mathbf{y}}}(-\log(p_f(\hat{\mathbf{z}}; \Psi)) - \log(p_h(\hat{\mathbf{y}}; \hat{\mathbf{z}}, \Theta)))$ is known after decoding $\hat{\mathbf{y}}$ and $\hat{\mathbf{z}}$. We claim that there is a real number step size $\rho_h^*$ that decrease the reconstruction error such that;

$$d(\mathbf{x}, g_s(\hat{\mathbf{y}}; \theta)) \geq d(\mathbf{x}, g_s(\hat{\mathbf{y}} + \rho_h^* \nabla_{\hat{\mathbf{y}}}(-\log(p_f(\hat{\mathbf{z}}; \Psi)) - \log(p_h(\hat{\mathbf{y}}; \hat{\mathbf{z}}, \Theta))); \theta)).$$

$\rho_h^*$ can be found by searching out of handful candidates or any optimization method such that;

$$\rho_h^* = \arg\min_{\rho_h}(d(\mathbf{x}, g_s(\hat{\mathbf{y}} + \rho_h \nabla_{\hat{\mathbf{y}}}(-\log(p_f(\hat{\mathbf{z}}; \Psi)) - \log(p_h(\hat{\mathbf{y}}; \hat{\mathbf{z}}, \Theta))); \theta))). \tag{8}$$

Our proposal can be seen as shifting the side latent by $\hat{\mathbf{z}} \leftarrow \hat{\mathbf{z}} + \rho_f^* \nabla_{\hat{\mathbf{z}}}(-\log(p_f(\hat{\mathbf{z}}; \Psi)))$ after decoding $\hat{\mathbf{z}}$ and shifting the main latent by $\hat{\mathbf{y}} \leftarrow \hat{\mathbf{y}} + \rho_h^* \nabla_{\hat{\mathbf{y}}}(-\log(p_f(\hat{\mathbf{z}}; \Psi)) - \log(p_h(\hat{\mathbf{y}}; \hat{\mathbf{z}}, \Theta)))$[5] after decoding $\hat{\mathbf{y}}$ while the best step sizes $\rho_f^*, \rho_h^* \in \mathbb{R}$ are to be found at encoding time and added to the bitstream with fixed bitlength.

## 5 Experimental Results

In this section, we show the main results, the complexity analysis and some important ablation studies.

### 5.1 Main Results

We used the CompressAI library (Bégaint et al., 2020) to test our contributions on 4 pre-trained neural image codecs named **bmshj2018-factorized** in Ballé et al. (2017), **mbt2018-mean** and **mbt2018** in Minnen et al. (2018), **cheng2020-attn** in Cheng et al. (2020), and 2 additional pre-trained codec named **invcompress** in Xie et al. (2021) and **DCVC-intra** in Li et al. (2022) with all intra mode and two neural video codec named **SSF** in Agustsson et al. (2020) and **DCVC** in Li et al. (2022). For the evaluation, we used the Kodak dataset (Kodak) and Clic-2021 Challenge's Professional dataset (CLI). The codecs are taken off the shelf and are not retrained. The rate was calculated from the final length of the compressed data and RGB PSNR is used for distortion. To evaluate performance, we utilize the bd-rate metric (Bjontegaard, 2001), which measures the bitrate savings achieved for an overlapped range of distortion level (Karczewicz & Ye, 2022). We conduct our evaluation on 6 to 8 pre-trained models provided by compressAI and the authors official implementation, whose bitrates range from 0.1bpp to 1.6bpp for image codecs and 0.03bpp to 0.35bpp for video codecs. We report our uniform VQ proposals under the name of **Hex-Quant** for regular hexagonal. We denote our gradient based improvement by **Latent Shift** and the results using both methods as **Join**.

Table 1: Average BD-Rate gains of our proposals for different baseline image codecs on 2 image datasets.

| Baseline Codec | Kodak Test Set | | | Clic-2021 Test Set | | |
|---|---|---|---|---|---|---|
| | Hex-Quant. | Latent Shift | Join | Hex-Quant. | Latent Shift | Join |
| bmshj2018-factorized | -0.62% | -0.49% | -1.08% | -1.78% | -0.69% | -2.26% |
| mbt2018-mean | -0.88% | -1.27% | -2.24% | -0.76% | -1.21% | -2.03% |
| mbt2018 | -0.55% | -1.44% | -1.98% | -0.66% | -1.71% | -2.33% |
| cheng2020-attn | -0.16% | -0.46% | -0.73% | -0.32% | -0.72% | -1.15% |
| InvCompress | -0.21% | -0.55% | -0.82% | -0.52% | -0.63% | -1.12% |
| DCVC-intra mode | -0.97% | -0.30% | -1.28% | -1.22% | -0.11% | -1.44% |

**Results on Image Codecs:** In Table 1, we present the results obtained by our methods on different datasets and codecs. Since our two proposals are orthogonal to each other, the gain of the **Join** method is almost the sum of the two. However, after uniform VQ, the reconstructed latents change compare to uniform SQ, thus, gradients wrt this latents change as well. This variation creates slightly different results compared to **Latent Shift** only, thus **Join** results are not the exact sum. Since uniform VQ's reconstructed latent is much more closer to the latents before quantization, nearly in all cases, **Latent Shift** with uniform VQ gives better contribution than only **Latent Shift**.

---

[5]Since we do not find significantly different correlation between the gradient of the main information's entropy and the gradient of the reconstruction error, we apply it by $\hat{\mathbf{y}} \leftarrow \hat{\mathbf{y}} + \rho_h^* \nabla_{\hat{\mathbf{y}}}(-\log(p_h(\hat{\mathbf{y}}; \hat{\mathbf{z}}, \Theta)))$ in our tests.

Figure 5a presents the bd-rate (gain in %) compared to **mbt2018-mean** for different quality levels on the Kodak dataset. Even though our proposal has a 2.24% gain, the gain on lower quality (thus lower rate) is bigger (3.5%). More detailed results regarding the gain w.r.t. to reconstruction quality are provided in Appendix G. According to our tests, since the side latent's correlation's is weaker, the most important gain of **Latent Shift** comes from the gradient of the main latent. The analysis of the source of the gain of **Latent Shift** and its performance against some alternatives are provided in 5.3.

Table 2: Average BD-Rate of our proposals for SSF and DCVC video codecs on UVG dataset and Bunny video in low-delay configuration. **Join** refers using **Oct-Quant** and **Latent Shift**.

| | SSF | | | | DCVC | | | |
| --- | --- | --- | --- | --- | --- | --- | --- | --- |
| Video | Hex-Quant. | Oct-Quant. | Latent Shift | Join | Hex-Quant. | Oct-Quant. | Latent Shift | Join |
| Beauty | -1.31% | -1.60% | -0.13% | -1.76% | -1.92% | -1.97% | -0.10% | -2.02% |
| Bosphorus | -1.88% | -2.29% | -0.70% | -3.23% | -1.23% | -1.48% | -0.62% | -2.05% |
| Honeybee | -1.75% | -1.10% | -0.88% | -2.00% | -1.80% | -2.13% | -0.56% | -2.65% |
| Jockey | -1.04% | -1.48% | -0.06% | -1.57% | -1.42% | -1.32% | -0.11% | -1.51% |
| ReadySteadGo | -1.03% | -2.00% | -1.52% | -3.74% | -1.29% | -1.30% | -1.08% | -2.35% |
| ShakeNDry | -1.38% | -2.83% | -0.35% | -3.25% | -1.55% | -1.65% | -0.23% | -1.92% |
| YatchRide | -1.03% | -1.67% | -0.14% | -1.87% | -1.42% | -1.61% | -0.18% | -1.83% |
| UVG Average | -1.31% | -1.99% | -0.60% | -2.70% | -1.52% | -1.64% | -0.41% | -2.05% |
| Bunny | -1.12% | -1.87% | -1.31% | -3.24% | -0.90% | -1.02% | -1.21% | -2.25% |

**Results on Video Codec:** In Table 2, we show the results of the **SSF** and **DCVC** video codecs for different sequences. The **SSF** (Agustsson et al., 2020) was proposed for low-delay mode: frames are coded sequentially using the previously reconstructed frame and an intra frame is inserted every 8 frames (UVG videos are divided into 75 Intra periods). I frame encoding uses the same VAE method, so our implementation on I frame is the same as the one described above. However, each P frame consists of motion information and residual information encoded by two different VAE models. Since the motion information represents only a fraction of the total bitrate ($3 - 4\%$), we apply our proposal only on the residual coding only. When we apply **Oct-Quant** and **Latent Shift** together, it yields a bitrate gain of $2 - 2.7\%$ for the same quality on UVG dataset, on average. Again, the gain at lower rate is larger (more than 4%) as demonstrated in Figure 5b. More analysis on quantization effects can be found in Appendix D.

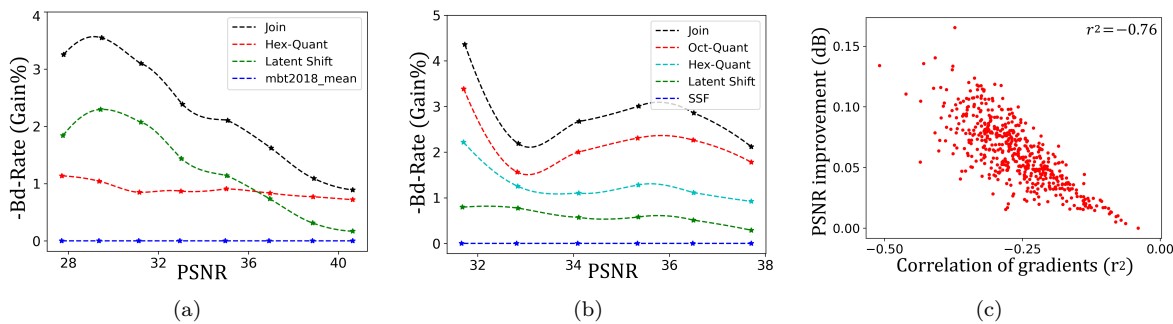

(a)        (b)        (c)

Figure 5: BD-Rates of our proposals from baseline codecs for different quality. a) mbt2018mean image codec on Kodak test set b) SSF video codec on UVG test set. c) Correlation between improvement on reconstruction quality and correlation of gradients on Clic and Kodak datasets.

**Gradients correlation:** The **Latent Shift**'s gain depends on how the gradients are correlated. To assess this correlation, we show the scatter plot of the individual gains in dB achieved by **Latent Shift** and the actual correlation coefficient between gradients for all test images and for all quality level in Figure 5c. We observed a strong negative correlation of $-0.76$ where this correlation is independent of the datasets, reconstruction quality or model. Since the neural video codec was optimized for the loss of all frames in a GOP, the gradients' sum in equation 3 and equation 4 are zero in expectation for both the frames in the

GOP and over the datasets. This smooths the sum of gradients even further and decrease the correlations compare to the image codecs. It explains why the the correlation of the gradients are lower and also why **Latent Shift**'s individual gain is lower in video codecs.

## 5.2 Complexity Analysis

Since the time complexity without parallelization is a good proxy of energy consumption, we performed the test on a single core CPU with preventing multi-thread operations. We measured the computational time of **cheng2020_attn**, **mbt2018-mean** on Kodak dataset and **SSF** on bunny dataset. The results in Table 3 demonstrate the relative encoding/decoding time of our proposed models, namely **Hex-Quant**, **Oct-Quant**, and **Latent Shift**, compared to the baseline models.

| Model | Encoding | | | Decoding | | |
|---|---|---|---|---|---|---|
| | Hex Quant | Oct Quant | Latent Shift | Hex Quant | Oct Quant | Latent Shift |
| mbt2018-mean | +3.0% | +5.0% | x10.1 | +1.8% | +1.8% | +0.70% |
| cheng2020-attn | +0.7% | +1.1% | x5.3 | +0.1% | +0.1% | +0.06% |
| SSF | +2.4% | +3.2% | x6.0 | +1.9% | +1.9% | +0.07% |

Table 3: Encoding and decoding runtime complexity of our proposed methods compare to the baselines.

The main source of additional encoding complexity for **Hex-Quant** and **Oct-Quant** is finding the closest codebook. Naive search can result in up to 20% overhead complexity for a 3-dimensional VQ. However, there exists algorithms that can efficiently find the nearest quanta center for hexagonal and truncated octahedron grids as defined in (Agrell et al., 2002; Conway & Sloane, 1982). We adapted this solution to our model, reducing the additional encoding time to 3-5% for the **mbt2018-mean** model. Although **cheng2020-attn** and **mbt2018-mean** have similar absolute extra computation costs, the relative extra cost of **cheng2020-attn** is lower (1.1-0.7%) due to the high complexity of the baseline model. For SSF, we neglected the I-frame compression runtime (since it is the same as **mbt2018-mean**) but measured the average runtime for encoding and decoding of P-frames, which are done by two VAEs: one for motion information and another for residual information. The results of the relative encoding/decoding time for our **Hex-Quant**, **Oct-Quant**, and **Latent Shift** compared to the baseline models can be seen in Table 3. Given that the differences in the dimensions of VQ are only reflected in the reshaping of tensors and the number of indices to be retrieved from the dictionary during decoding, the extra decoding complexity between **Hex-Quant** and **Oct-Quant** is negligible. This is almost the same for all three models, as shown in Table 3.

The extra complexity in decoding introduced by the **Latent Shift** operation is due to the calculation of the gradient of entropy with respect to the latents and the shifting operation. Nonetheless, this extra complexity is negligible, accounting for less than 1% of the total decoding time, as shown in Table 3. In contrast, the encoding complexity of **Latent Shift** is much higher, ranging from 5 to 10 orders of magnitude higher than the baseline encoding time. This is because of the search for the best step size out of 8 candidates during encoding, which could be eliminated by using a single universal step size for all images. This solution has been implemented in the **Latent Shift** extension to traditional codecs, as described in Section 5.4.

## 5.3 Latent Shift versus Alternatives

In order to show how the proposed **Latent Shift** is better than alternatives, we shift the latent in a random direction at encoding time such that $\hat{\mathbf{y}} \leftarrow \hat{\mathbf{y}} + \epsilon(\rho_h)$, where $\epsilon(\rho_h) \in \mathbb{R}^{m \times m \times o} \sim \mathcal{N}(0, 1)$ and $\rho_h$ is a random seed to be signalled to the decoder. The best random seed and gradients, in terms of PSNR improvement, should be found at encoding time. We generate 1024 random gradient and encode the best random seed with 10 bits as an extra information. Even though this approach is costly in terms of computational complexity in encoding time, (it needs 1024 times forward pass), we think this is a natural baseline to our proposal, and we refer to it as **Random Shift**. Another alternative would be to use a constant gradient for all latents in a given image. Thus, at encoding time we need to test a large set of values and assume all latents should be shifted by this amount such that $\hat{\mathbf{y}} \leftarrow \hat{\mathbf{y}} + \rho_h$ where $\rho_h \in \mathbb{R}$. The best value of $\rho_h$ should be signaled to the decoder with 10 bit extra cost. We refer to this as **Scalar Shift** approach.

Our final alternative does not utilize the true gradient with respect to entropy directly, but instead relies on an approximation. Specifically, we make the assumption that the rate of change of any latent is independent and increases linearly by the distance from the center as $R(\hat{\mathbf{y}}) = a|\hat{\mathbf{y}} - \boldsymbol{\mu}| + b$ where $a, b \in \mathbb{R}^+$. Under this assumption, the magnitude of the gradient will be constant, but the direction will always point in the opposite direction to the center. Thus, this approach shifts the latent in the opposite direction of the distribution's center such that $\hat{\mathbf{y}} \leftarrow \hat{\mathbf{y}} - \rho_h \operatorname{sign}(\hat{\mathbf{y}} - \boldsymbol{\mu})$. In hyperprior entropy models, the latent is assumed to follow a Gaussian distribution, thus the latent's entropy gets smaller if it moves towards the center of the distribution. By shifting the latent in the opposite of this direction, the entropy increases. Since the factorized entropy model does not use Gaussian distribution, we shift the latent to the opposite direction of zero center means assuming $\boldsymbol{\mu} = 0$ in the factorized entropy. The best value of $\rho_h$ should be signaled to the decoder with 10 bit extra cost. We refer to this approach as **Sign Shift**.

Table 4: Average BD-PSNR of **Latent Shift** and some alternatives.

| Baseline Codec | Random Shift | Scalar Shift | Sign Shift | Latent Shift |
|---|---|---|---|---|
| bmshj2018-factorized | 0.0002 dB | 0.0036 dB | 0.0151 dB | 0.0297 dB |
| mbt2018-mean | 0.0006 dB | 0.0007 dB | 0.0339 dB | 0.0705 dB |

In Table 4, we present the results for **bmshj2018-factorized** (lowest correlation between gradients) and **mbt2018-mean** (highest correlation between gradients). Since all our alternatives need extra 10 bits signaling cost, we neglect it and assume that the bitlengths are the same as the baselines, and we report the BD-PSNR defined in Bjontegaard (2001). These results show that our proposal is significantly better than all our alternatives. The closest one, **Sign Shift**, reaches half the performance of **Latent Shift**. The random alternative could not improve the baseline significantly, even though their encoder is almost 1000 times computational demanding.

## 5.4 Enhancing Traditional Codecs with Latent Shift

Despite traditional codecs do not use gradient-based optimization methods and instead rely on discrete search algorithms within a restricted search space to minimize RD cost, we can still argue that the KKT condition exists. This is because the discrete search algorithms used by traditional codecs find the optimum transform coefficient of residuals (known as latent variables in neural compression literature) for both rate and distortion objectives.

In order to implement **Latent Shift** on the current traditional Sota codec **ECM.8.0** described in Coban et al. (2023), we need an entropy model to calculate its gradient with respect to the transform coefficients. However, since **ECM.8.0** uses a bit-level adaptive entropy model (CABAC), it is not possible to obtain a closed-form solution for the gradient. This may introduce additional complexity if we try to obtain the gradient by using finite difference methods. To avoid this complexity, we use a very simple approximation of the entropy model where the assumption is that the rate linearly increases with the absolute value of the coefficient. This enables us to express the gradient of the rate as **Sign Shift**, defined in 5.3. To avoid adding complexity to the encoder in **Latent Shift**, we set a universal step size, which we fine-tuned on the training set. This step size is then hard-coded into both the encoder and decoder devices in advance.

| Sequences | All Intra (Image compression) | | | | | Random Access (Video Compression) | | | | |
|---|---|---|---|---|---|---|---|---|---|---|
| | Y | U | V | EncT | DecT | Y | U | V | EncT | DecT |
| Class A1 | -0.11% | -0.11% | -0.11% | 100% | 100% | -0.07% | -0.04% | -0.16% | 100% | 100% |
| Class A2 | -0.07% | -0.10% | -0.04% | 100% | 100% | -0.04% | -0.35% | -0.05% | 100% | 100% |
| Class B | -0.10% | -0.10% | -0.14% | 100% | 100% | -0.07% | -0.20% | -0.15% | 100% | 100% |
| Class C | -0.07% | -0.16% | -0.29% | 100% | 100% | -0.07% | -0.29% | -0.22% | 100% | 100% |
| Class E | -0.09% | -0.14% | -0.08% | 100% | 100% | | | | | |
| Overall | -0.08% | -0.12% | -0.13% | 100% | 100% | -0.06% | -0.22% | -0.35% | 100% | 100% |

Table 5: Bd-rate of **Latent Shift** method for Y, U and V channel and relative encoding and decoding time percentage compare to ECM-8.0 on All Intra and Random Access mode under Common Test Conditions.

The results presented in Table 5 were obtained under the Common Test Conditions (CTC) (Karczewicz & Ye, 2022). According to the results, **Latent Shift** managed to save less than 0.1% on the Luma channel and more than 0.1% on the chroma channels in all intra mode settings where each frame is encoded separately. In Random Access mode, which is the most effective video compression setting, the results were similar. Thanks to the universal step size and the approximation of the gradient, our proposal has no impact on the encoding and decoding times, as reported in the Table 5.

### 5.5 Latent Shift after Finetuning Solutions

To demonstrate the orthogonality between **Latent Shift** and an encoder-side fine-tuning solution proposed in Campos et al. (2019), we conducted tests on two baseline models: one with and one without fine-tuning. In the fine-tuning approach, the latents are fine-tuned for 1000 iterations during encoding time, this incurs 1000 forward passes and 1000 backward gradient calculations, resulting in around 4000 times more complexity without parallelization. In contrast, our **Latent Shift** incurs negligible extra encoding time compared to fine-tuning solutions where the complexity can be found in Table 6. An important result can be observed when applying **Latent Shift** after fine-tuning solutions, as shown in Table 6. Fine-tuning solutions minimize the loss without averaging images in the train set, which strengthens the KKT conditions and increases correlations between gradients. For instance, the average gradients correlation for the Kodak dataset becomes $-0.2212$ after fine-tuning, compared to $-0.1805$ in the **mbt2018-mean** codec. The improved correlations lead to improved performance, as shown by the fact that the fine-tuning solution only achieves a rate saving of $-5.77\%$, while combining it with **Latent Shift** increases the saving to $-7.47\%$ for the Kodak dataset.

Table 6: **Latent Shift** performance over Finetuning solutions

| Model | EncT | DecT | Bd-Rate (Kodak) | Corr. (Kodak) | Bd-Rate (Clic) | Corr. (Clic) |
|---|---|---|---|---|---|---|
| bmshj2018 baseline | x1 | +0.0% | 0.0% | | 0.0% | |
| Only **Latent Shift** | x12 | +0.7% | -0.49% | -0.108 | -0.69% | -0.0952 |
| Only FineTuning | x4800 | +0.0% | -6.52% | | -6.73% | |
| FineTuning + **Latent Shift** | x4812 | +0.7% | -7.88% | -0.165 | -8.06% | -0.1578 |
| mbt2018-mean baseline | x1 | +0.0% | 0.0% | | 0.0% | |
| Only **Latent Shift** | x10.1 | +0.7% | -1.27% | -0.1805 | -1.21% | -0.1465 |
| Only FineTuning | x4380 | +0.0% | -5.77% | | -5.53% | |
| FineTuning + **Latent Shift** | x4390 | +0.7% | -7.47% | -0.2212 | -7.16% | -0.1796 |

## 6 Conclusion

In this work, we have proposed two orthogonal methods to improve further the latent representation of generic compressive variational auto-encoders (VAE). Firstly, we exploit further the remaining redundancy in the latent during the quantization stage. To do so, we demonstrated that a uniform VQ method improves a VAE trained using uniform scalar quantization. Secondly, we exploit the correlation between the gradient of the entropy and the reconstruction error to improve the latent representation. The combination of these two methods improve the latent representation, and bring significant gains on top of several state-of-the-art compressive auto-encoders, without any need of retraining. From these results, several improvements can be foreseen. First, the correlation of the gradients depends on the training set according to the definition of the KKT conditions. However, even though different models use the same training set and training procedure, their gradients' correlation coefficient may be different. Thus, it might be interesting to explore the connection of certain type of model architectures with the correlation of the gradients. Secondly, the uniform VQ process was applied without retraining. Even though VQ quantization error is lower on average, the maximum quantization error can be higher. Since the decoder block cannot see these individual high errors during the training, the model becomes sub-optimal. This can be solved by re-training the decoder block or the entire model using a continuous relaxation of the uniform VQ grid.

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

## A    Proof of Theorem 1

*Proof.* Scalar quantization map can be see as $n+1$ border values with $b_0 < b_1 < \cdots < b_n$ and $n$ quantization center with $c_1 < c_2 < \cdots < c_n$. Any latent $y$ with $b_{i-1} \leq y < b_i$ should be quantized to the point $c_i$. Quantization map can be represented by an ordered set consisting of all borders and centers such as $\mathbb{M} = \{b_0 < c_1 < b_1 < c_2 < b_2 < \cdots < c_n < b_n\}$ where $|\mathbb{M}| = 2n + 1$. When the quantization map is non-uniform, the difference between consecutive elements in the set are not necessarily equal, i.e. $\exists (i,j), \mathbb{M}_i - \mathbb{M}_{i-1} \neq \mathbb{M}_j - \mathbb{M}_{j-1}$. Nearest integer quantization map can be denoted as $\mathbb{M}^{(u)} = \{0.5, 1, 1.5, 2, 2.5, \ldots n, n+0.5\}$ or simply $\mathbb{M}_i^{(u)} = 0.5i$ thus,$\forall i, \mathbb{M}_i^{(u)} - \mathbb{M}_{i-1}^{(u)} = 0.5$. Let's assume than any arbitrary $\mathbb{M}$ is the optimal scalar quantization map of a neural codec's latent $y$ obtained by $y = g_a(x)$. Since $\mathbb{M}$ and $\mathbb{M}^{(u)}$ are both monotonic increasing set, there exists a bijective function $f(.)$ that maps $\forall i, \mathbb{M}_i$ to $\mathbb{M}_i^{(u)}$, and $f^{-1}(.)$ maps $M_i^{(u)}$ to $\mathbb{M}_i$, $\forall i$. According to universality theorem Hornik et al. (1989), the function $f(.)$ can be implemented by a multi layer neural network.

Two codecs' performances are equal if and only if entropy of their latents are equal and their reconstructions are the same. First, we start by showing that their entropy's are the same by showing that the corresponding center's PMF are equal in both space, i.e. $\forall i, P(i) = P^{(u)}(i)$. We can write the $i$-th quantization center's PMF as $P(i) = \int_{b_{i-1}}^{b_i} p(y)dy$ where $y$ is a point in $g_a$'s output space. Any point $y$ can be transformed to a new space by $z = f(y)$. We can write the $i$-th quantization center's PMF as $P^{(u)}(i) = \int_{i-0.5}^{i+0.5} p(z)dz$ in this space. We can rewrite it as $P^{(u)}(i) = \int_{f^{-1}(i-0.5)}^{f^{-1}(i+0.5)} p(f^{-1}(z))df^{-1}(z)$. Since $f^{-1}(i+0.5) = b_i$, $f^{-1}(i-0.5) = b_{i-1}$ and $f^{-1}(z) = y$, we can write $\forall i, P^{(u)}(i) = \int_{b_{i-1}}^{b_i} p(y)dy = P(i)$.

The output of a deep decoder is $g_s(c_i)$ if the latent $y = g_a(x)$ meets $b_{i-1} \leq y < b_i$. Any latent $b_{i-1} \leq y < b_i$ is mapped to $f(b_{i-1}) \leq f(y) < f(b_i)$ thus $i - 0.5 \leq z < i + 0.5$. The latent $z$ which lies between $i - 0.5$ to $i + 0.5$, can be quantized to $i$ in the new space. Since the decoder applies $g_s(f^{-1}(z))$, its output should be $g_s(f^{-1}(i))$. Since $f^{-1}(i) = c_i$, it gives $g_s(c_i)$. □

## B    Advantage of Space Tessellation Grid on Quantization

When the source has a uniform distribution, quantization using a truncated octahedron gives better RD performance compared to regular hexagonal grids, and regular hexagonal grid gives better RD performance than uniform SQ grids (nearest integer rounding). In order to show the superiority of a method over another in terms of RD performance, it is enough to compare the reconstruction error at equal bit-rate. Since the equal volume grids have the same probability under uniform distribution, the rate is equal for the three cases. Since the distributions are identical, we just need to compute the mean square error for each types of grid at some position, for example the origin for 1D, 2D and 3D cases respectively.

MSE of uniform SQ grid can be written as the integral of square error normalized by the grid size:

$$MSE^{(s)}(u) = \frac{1}{u} \int_{-u/2}^{u/2} x^2 dx. \tag{9}$$

It gives $MSE^{(s)}(u) = u^2/12 \approx 0.0833u^2$.

**Hexagonal grid case:** For hexagonal grid, we need to double integrate over the hexagonal domain. Figure 6a shows an hexagon with a side a length $a$ located at the origin. We also show each side's functions in 2D space. We divide the hexagon into two parts for positive and negative $y$ and calculate the analytic integral for these two region separately. We then normalize the sum of squares by area of the area of the hexagon, which is $3\sqrt{3}a^2/2$.

$$MSE^{(h)}(a) = \frac{2}{3\sqrt{3}a^2} \left( \int_0^{a\frac{\sqrt{3}}{2}} \int_{-a+\frac{y}{\sqrt{3}}}^{a-\frac{y}{\sqrt{3}}} \frac{x^2 + y^2}{2} dxdy + \int_{-a\frac{\sqrt{3}}{2}}^{0} \int_{-a-\frac{y}{\sqrt{3}}}^{a+\frac{y}{\sqrt{3}}} \frac{x^2 + y^2}{2} dxdy \right). \tag{10}$$

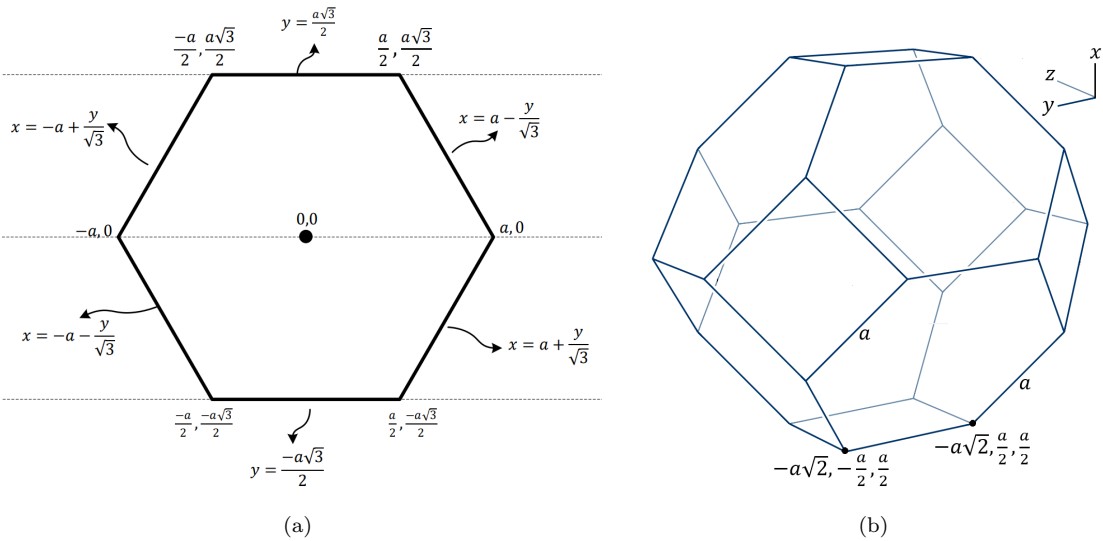

Figure 6: a) Regular hexagon and b) Truncated octahedron located at origin.

When we first integrate the positive part of integral over $x$ between $-a + \frac{y}{\sqrt{3}}$ and $a - \frac{y}{\sqrt{3}}$, followed by the integral over $y$ between 0 and $a\frac{\sqrt{3}}{2}$. The integral for the positive region is is then $\frac{5\sqrt{3}a^4}{32}$. The same can be done for the negative region, but since the distribution is uniform, it gives the exact same result, thus we conclude that $MSE^{(h)}(a) = \frac{5a^2}{24}$. In order to obtain the same volume for the grid, hexagon's area should be $u^2$. Since the hexagon's area is $3\sqrt{3}a^2/2$ for a side length if $a$, we find that $a = \frac{\sqrt{2}u}{\sqrt{3\sqrt{3}}}$. Thus

$$MSE^{(h)}(u) = \frac{5\sqrt{3}u^2}{108} \approx 0.0801u^2.$$

**Truncated octahedron case:** For the MSE of the truncated octahedron grid, we need to integrate the mean square error of each 3 dimension over the truncated octahedron domain. In figure 6b, a truncated octahedron with a side length of $a$ is shown. The solution can be obtained by integrating one octant, multiply it by 8 and normalize by its volume which is $8\sqrt{2}a^3$ as follows:

$$MSE^{(o)}(a) = \frac{8}{8\sqrt{2}a^3} \int_0^{\sqrt{2}a} \int_0^{min(\sqrt{2}a, 3\sqrt{2}a/2 - x)} \int_0^{min(\sqrt{2}a, 3\sqrt{2}a/2 - x - y)} \left( \frac{x^2 + y^2 + z^2}{3} \right) dz\,dy\,dx \quad (11)$$

First, we integrate over $z$, $y$ and then $x$, which gives the solution $MSE^{(o)}(a) = \frac{19}{48}a^2$. To compare with the same grid volume, the truncated octahedron should have a volume of $u^3$. Since the volume is $8\sqrt{2}a^3$ when one side length is $a$, we can find that $a = \frac{u}{(8\sqrt{2})^{1/3}}$, thus $MSE^{(o)}(u) = \frac{19}{48(8\sqrt{2})^{2/3}}u^2 \approx 0.0785u^2$.

As a result, $MSE^{(s)} \approx 0.0833u^2$, $MSE^{(h)} \approx 0.0801u^2$ and $MSE^{(o)} \approx 0.0785u^2$, where the volume of the grid is $u$. We can thus conclude that $\forall u \in \mathbb{R}^+, MSE^{(o)}(u) < MSE^{(h)}(u) < MSE^{(s)}(u)$.

## C  Proof of Theorem 2 and Corollary 2.1

In this section, we show the proofs of Theorem 2 and Corollary 2.1.

### C.1  Theorem 2

*Proof.* Since $\hat{\mathbf{z}}$ is a function of $\phi, \Phi$ and $\mathbf{x}$, we can define a function for the first objective of equation 1 by $\mathcal{L}_1(\mathbf{x}, \phi, \Phi, \Psi) := -\log(p_f(\hat{\mathbf{z}}; \Psi))$, and $\hat{\mathbf{y}}$ is a function of $\phi$ and $\mathbf{x}$ we can define a function for the second objective of equation 1 by $\mathcal{L}_2(\mathbf{x}, \phi, \Phi, \Theta) := -\log(p_h(\hat{\mathbf{y}}; \hat{\mathbf{z}}, \Theta))$. For the third objective of equation 1, we can define a function by $\mathcal{L}_3(\mathbf{x}, \phi, \theta) := d(\mathbf{x}, g_s(\hat{\mathbf{y}}; \theta))$. When considering the coefficient of the objectives are defined by $\alpha_1 := 1/(2 + \lambda)$, $\alpha_2 := 1/(2 + \lambda)$, and $\alpha_3 := \lambda/(2 + \lambda)$, we can write equation 1 as follows.

$$\phi^*, \theta^*, \Phi^*, \Theta^* \Psi^* = \underset{\phi, \theta, \Phi, \Theta, \Psi}{\arg\min} \left( \mathbb{E}_{\mathbf{x} \sim p_x} \left[ \alpha_1 \mathcal{L}_1(\mathbf{x}, \phi, \Phi, \Psi) + \alpha_2 \mathcal{L}_2(\mathbf{x}, \phi, \Phi, \Theta) + \alpha_3 \mathcal{L}_3(\mathbf{x}, \phi, \theta) \right] \right) \quad (12)$$

Since $\lambda > 0, \forall i, \alpha_i > 0$ and $\sum_i \alpha_i = 1$, the set of $\alpha_i$s corresponds to coefficients set and $\mathcal{L}_i(.)$s corresponds to objectives, equation 12 shows an unconstrained multi-objective optimization problem. This problem has five set of variables to be optimized and gradients w.r.t each variable set should meet KKT conditions. Since $\mathcal{L}_3(\mathbf{x}, \phi, \theta)$ does not depends on $\Phi$, thus, $\nabla_\Phi \mathcal{L}_3(\mathbf{x}, \phi, \theta) = 0$, we can write KKT conditions w.r.t $\Phi$ as follows.

$$\mathbb{E}_{\mathbf{x} \sim p_x} \left[ \alpha_1 \nabla_\Phi \mathcal{L}_1(\mathbf{x}, \phi, \Phi, \Psi) + \alpha_2 \nabla_\Phi \mathcal{L}_2(\mathbf{x}, \phi, \Phi, \Theta)) \right] = 0.$$

Since $\alpha_1 = \alpha_2$, we obtain the first condition in equation 3 by simply replacing $\mathcal{L}_1(.)$ and $\mathcal{L}_2(.)$ with their definitions. Similarly, we can write KKT conditions w.r.t $\phi$ as follows.

$$\mathbb{E}_{\mathbf{x} \sim p_x} \left[ \alpha_1 \nabla_\phi \mathcal{L}_1(\mathbf{x}, \phi, \Phi, \Psi) + \alpha_2 \nabla_\phi \mathcal{L}_2(\mathbf{x}, \phi, \Phi, \Theta)) + \alpha_3 \nabla_\phi \mathcal{L}_3(\mathbf{x}, \phi, \theta) \right] = 0.$$

This is equivalent to the equation 4 when $\alpha_i$s and $\mathcal{L}_i(.)$s are replaced with their definitions.

$\square$

## C.2 Corollary 2.1

*Proof.* When we remove the expectation term from equation 3, we get followings.

$$\nabla_\Phi \log(p_f(\hat{\mathbf{z}}; \Psi)) = -\nabla_\Phi \log(p_h(\hat{\mathbf{y}}; \hat{\mathbf{z}}, \Theta)).$$

Since $\hat{\mathbf{z}}$ and $\Phi$ have dependency ($\hat{\mathbf{z}} = Q(h_a(\mathbf{y}; \Phi))$) partial derivations are not zero. Thus, if we multiply both side with jacobian matrix $\mathbb{J}^{(\Phi, \hat{\mathbf{z}})}$ where $\mathbb{J}_{i,j}^{(\Phi, \hat{\mathbf{z}})} = \frac{\partial \Phi_j}{\partial \hat{\mathbf{z}}_i}$, we change the gradient variable from $\Phi$ to $\hat{\mathbf{z}}$ and reach equation 5.

We can repeat the same steps for equation 6. When we remove the expectation term from equation 4, we get followings.

$$\nabla_\phi \left[ -\log(p_f(\hat{\mathbf{z}}; \Psi)) - \log(p_h(\hat{\mathbf{y}}; \hat{\mathbf{z}}, \Theta)) \right] = -\lambda \nabla_\phi d(\mathbf{x}, g_s(\hat{\mathbf{y}}; \theta))$$

Since $\hat{\mathbf{y}}$ and $\phi$ have dependency ($\hat{\mathbf{y}} = Q(g_a(\mathbf{x}; \phi))$) partial derivations are not zero. Thus, if we multiply both side with jacobian matrix $\mathbb{J}^{(\phi, \hat{\mathbf{y}})}$ where $\mathbb{J}_{i,j}^{(\phi, \hat{\mathbf{y}})} = \frac{\partial \phi_j}{\partial \hat{\mathbf{y}}_i}$, we change the gradient variable from $\phi$ to $\hat{\mathbf{y}}$ and reach equation 6.

$\square$

# D Ablation Studies

In this section, we present several ablation studies for uniform VQ and latent shifting.

## D.1 Gain Analysis of Latent Shift

Table 7: Upper limits of the gradient based latent shifting on **mbt2018-mean** codec.

|  | Only Side Shift (BD-Rate) | Only Main Shift (BD-Psnr) | Only Main Shift (BD-Rate) | Main & Side Shift (BD-Rate) |
|---|---|---|---|---|
| **True Gradients** | -1.011% | 1.3972 dB | -25.139% | -26.150% |
| **Latent Shift** | -0.031% | 0.0705 dB | -1.270% | -1.301% |

To understand the upper limits of gradient based latent shifting, we use the true gradients instead of proxy one and measure the performance. Simply, we shift the side latents by $\hat{\mathbf{z}} \leftarrow \hat{\mathbf{z}} + \rho_f^* \nabla_{\hat{\mathbf{z}}}(-log(p_h(\hat{\mathbf{y}}; \hat{\mathbf{z}}, \Theta)))$ after decoding $\hat{\mathbf{z}}$. Later, we shift the main latent by $\hat{\mathbf{y}} \leftarrow \hat{\mathbf{y}} + \rho_h^* \nabla_{\hat{\mathbf{y}}}(d(\mathbf{x}, g_s(\hat{\mathbf{y}}; \theta)))$ after decoding $\hat{\mathbf{y}}$. We can see this hypothetical case as if the correlations are $-1$. This case is mentioned by **True Gradients** in the results on Table 7 while our proposal is **Latent Shift**. The results are taken with **mbt2018-mean** image codec on Kodak dataset.

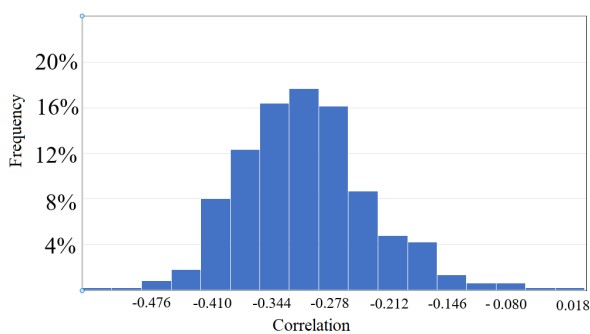

Figure 7: Histogram of correlation between gradients wrt main latents. The data is taken with **mbt2018-mean** image codec on Kodak and Clic dataset.

According to this results, we can see that even if the side latent's gradients were perfectly correlated, our maximum gain would be around 1%. Since the correlation of gradients wrt the side latent is weak ( $r^2 \approx -0.07$), our gain is negligible. As a consequence, in practice, shifting side latent may not be neglected. On the other hand, main latents true gradients increase PSNR by 1.40dB in average which is equivalent of saving around 25% of the bitstream. Our proposal could increase the PSNR by 0.07dB in average which is equivalent to saving 1.27% of the bitstream for the same quality thanks to the existing correlation between gradients wrt main latents as shown in Figure 7. The gain is of course smaller than the upper limit, but significant still. Since keeping these gradients are costly (nearly the same cost of saving image itself), searching more effective way of using those gradient is meaningful.

## D.2 Truncated Octahedron versus Hexagon

Concerning uniform VQ quantizations, either hexagonal grid quantization **Hex-Quant** or truncated octahedron grid quantization **Hex-Quant** give different results. Better quantization leads to the smaller quantization error on latent and better reconstruction. However, bigger symbol dictionary is needed (square of SQ's dictionary size in hexagonal grid). Since the arithmetic encoder has a fixed bit resolution, it has representation limitations (for 16-bit resolution, the minimum probability is 1/65536). Thus, in practice, we can assign 1/65536 probability to the symbol whose probability is lower than 1/65536, what makes encoding less efficient and increase the rate. Another alternative is to remove those symbols, what increases the quantization error of latents but decreases the rate. In practice, we have chosen to remove the symbol if its probability is less than $10^{-7}$, and we found out that it always gives better RD performance.

In this ablation study, we analyse further the quantization effects on reconstruction error (PSNR), as well as the impact of latent quantization error in terms of Signal Noise Ration (SNR) w.r.t. the rate (under the assumption that arithmetic encoder has infinite bit resolution). To this end, we do not encode the symbols into bitstream, but calculate the lower bound of bitlength according to Shanon's entropy theorem without limits of integer resolution that dictates some certain probabilities on symbols. We calculate SNR of latents by $SNR = -10log_{10}(d(\hat{\mathbf{y}}, \mathbf{y}))$ where $\mathbf{y}$ is latent, $\hat{\mathbf{y}}$ is reconstructed latent by certain quantization techniques and $d(.,.)$ measures MSE error between two inputs.

We use **mbt2018-mean** neural codec in Minnen et al. (2018) on Kodak dataset. Results are presented in Table 8 and show that **Oct-Quant** gives lower reconstruction error (higher PSNR improvement) and lower quantization error over latent (higher SNR improvement) than **Hex-Quant** for the same rate. In addition, **Oct-Quant** saves more bitlength than **Hex-Quant** for the same reconstruction quality. Figure 8 compares the performances of **Oct-Quant** and **Hex-Quant** to the uniform SQ baseline, for different rate and reconstruction quality.

Table 8: Performance of **Hex-Quant** and **Oct-Quant** compare to uniform SQ.

| Quantization | BD-PSNR | BD-SNR | BD-Rate |
|---|---|---|---|
| Hex-Quant | 0.0374 dB | 0.0600 dB | -0.748% |
| Oct-Quant | 0.0480 dB | 0.0736 dB | -0.957% |

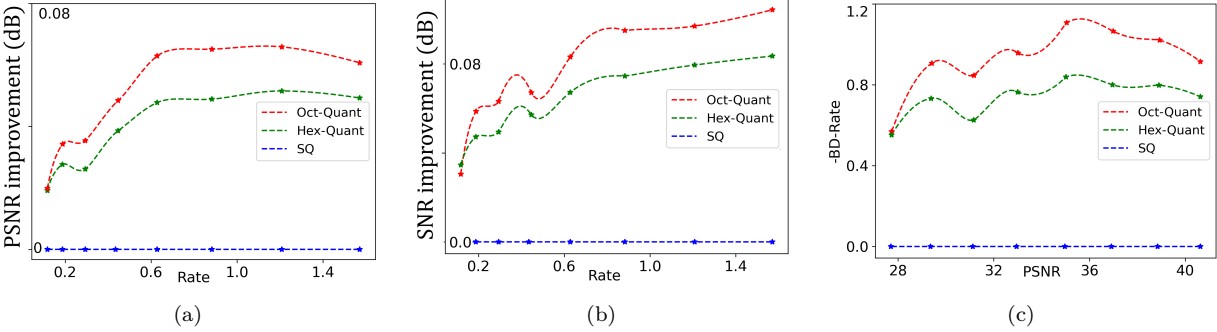

(a)            (b)            (c)

Figure 8: BD-Psnr, BD-Snr and -Bd-Rate performance of Hex-Quant and Oct-Quant compare to the uniform SQ. In all perspective, Oct-Quant is the best, while Hex-Quant comes for the second.

# E   Numeric PMF Calculation for Hexagonal Domain

In this section, we give some implementation details.

The closed form solution for the integrals of multidimensional known densities over any domain has very specific solutions, and generally no tractable form (Savaux & Le Magoarou, 2020). That holds even for Gaussian distribution of dimension 2 on the regular hexagon domain. Thus, there is no closed form solution of equation 2, where the probabilities are independent Gaussian on the domain of regular hexagonal. However, we find the solution using the combination of both analytic and numerical integration as detailed below.

The integral centred around $(G_x, G_y)$ of hexagonal domain $G$ for the independent Gaussian distribution is given by

$$P(G_x, G_y) = \int_G N(x, \mu_1, \sigma_1).N(y, \mu_2, \sigma_2).dx.dy. \tag{13}$$

For sake of simplicity, let our grid is located on the center $(G_x, G_y) = (0, 0)$ and $a$ be the one side length of the hexagonal. Then, the integral is defined as

$$P(0,0) = \int_0^{a\frac{\sqrt{3}}{2}} \int_{-a+\frac{y}{\sqrt{3}}}^{a-\frac{y}{\sqrt{3}}} N(x, \mu_1, \sigma_1).N(y, \mu_2, \sigma_2).dx.dy+$$

$$\int_{-a\frac{\sqrt{3}}{2}}^{0} \int_{-a-\frac{y}{\sqrt{3}}}^{a+\frac{y}{\sqrt{3}}} N(x, \mu_1, \sigma_1).N(y, \mu_2, \sigma_2).dx.dy, \tag{14}$$

where the hexagon is divided into lower and upper half parts, and we sum the two integrals. It is noted that the integral does not admit a closed form solution, but the inner integral has the closed form and outer integral does not have the analytic solution. The solution of the inner integral in the first part is given by

$$\int_{-a+\frac{y}{\sqrt{3}}}^{a-\frac{y}{\sqrt{3}}} N(x, \mu_1, \sigma_1).dx = \frac{1}{2}erf\left(\sqrt{2}\frac{(-\mu_1 + x)}{2\sigma_1}\right)\Big|_{x=-a-\frac{y}{\sqrt{3}}}^{x=a-\frac{y}{\sqrt{3}}} \tag{15}$$

by substituting into equation 14, we obtain:

$$\int_0^{a\frac{\sqrt{3}}{2}} \int_{-a+\frac{y}{\sqrt{3}}}^{a-\frac{y}{\sqrt{3}}} N(x,\mu_1,\sigma_1).N(y,\mu_2,\sigma_2).dx.dy$$

$$= \int_0^{a\frac{\sqrt{3}}{2}} \frac{1}{2} erf\left(\sqrt{2}\frac{(-\mu_1+x)}{2\sigma_1}\right) \Bigg|_{x=-a-\frac{y}{\sqrt{3}}}^{x=a-\frac{y}{\sqrt{3}}} N(y,\mu_2,\sigma_2).dy \quad (16)$$

This is finally solved by the numerical integration[6]. Similarly, we can also obtain the integral of the second part.

## F  Complexity Analysis

Here we describe the computational complexity of proposed method and provide detailed analysis of source of the additional computational costs of the proposals. The complexity of our proposed method includes encoding and decoding complexity of uniform VQ, and encoding and decoding complexity of Latent shift, and they are detailed below.

### F.1  Encoding complexity of uniform VQ

First, we should define our grid centers as codebook such that $\mathbf{c}^{(i)} \in R^v, i = 1\ldots M$ be the $M$ grid centers of the $v$-dimensional shape (e.g hexagonal grids center for $v = 2$) and PMF of our $v$-dimensional grid using learned $1D$ PMF by equation 2 in the main paper.

The additional steps involved in the encoding time over the baseline approach are as follows:

1. Reshape the latents into pseudo $v$-dimensional vector $\mathbf{y} \in R^{m \times m \times o} \to \check{\mathbf{y}} \in R^{b \times v}$, where $b = \frac{m.m.o}{v}$ (before reshaping, the latent can be sorted by their distribution's $\sigma$ parameters. In this way, the latents that belongs to a similar distribution can be dropped into the same latent vector).

2. Find closest codebook from our initial quantization grids for each $b$ vector such $\tilde{\mathbf{y}}_j = argmin_i || \check{\mathbf{y}}_j - \mathbf{c}^{(i)} ||$, where $\tilde{\mathbf{y}}_j \in \{1\ldots M\}, j = 1\ldots b$ are the codes to be encoded into bitstream.

In the shared step (with baseline method), $\tilde{\mathbf{y}}_j, j = 1\ldots b$ should be encoded into bitstream by given PMF. Thus, in encoding time above 2 step's complexity is the reason for the extra complexity introduced by our proposed method.

### F.2  Decoding complexity of uniform VQ

In decoding time, $\tilde{\mathbf{y}}_j, j = 1\ldots b$ should be decoded from bitstream by provided PMF table. The source of extra complexity of our method (over the baseline) in the decoding time is the following two steps:

1. De-quantizate the decoded codes: find the center of selected quanta center such that $\bar{\mathbf{y}}_j = \mathbf{c}^{\tilde{\mathbf{y}}_j}, j = 1\ldots b$

2. Reshape dequantized latent into the original dimensions such that $\bar{\mathbf{y}} \in R^{b \times v} \to \hat{\mathbf{y}} \in R^{m \times m \times o}$, (if the ordering wrt $\sigma$ is applied, we need to revert the order back in order to have the latents in original order)

In shared step (with the baseline method), these dequantized and reshaped latents are fed to the decoder network.

---

[6]https://github.com/esa/torchquad

### F.3 Encoding complexity of Latent Shift

In encoding time, the complexity introduced by the *Latent Shift* are as follows:

1. The calculation of the gradients of the entropy $\nabla_{\hat{\mathbf{y}}}(-log(p_h(\hat{\mathbf{y}}; \hat{\mathbf{z}}, \Theta)); \theta)$

2. finding the best step size $\rho_h$ that maximize the reconstruction quality when the latent is shifted as $\hat{\mathbf{y}} \leftarrow \hat{\mathbf{y}} + \rho_h^* \nabla_{\hat{\mathbf{y}}}(-log(p_h(\hat{\mathbf{y}}; \hat{\mathbf{z}}, \Theta)))$

It is noted that calculating gradient of the entropy has a negligible complexity, because we do not need the forward pass and there is closed form solution. When the entropy model uses Gaussian distribution such as $p_h(\hat{\mathbf{y}}; \hat{\mathbf{z}}, \Theta) := N(\hat{\mathbf{y}}; \mu, \sigma)$, the gradient of the entropy becomes the derivative of $-log(N(\hat{\mathbf{y}}; \mu, \sigma))$ wrt $\hat{\mathbf{y}}$ which has closed form solution, where $N(.; \mu, \sigma)$ is PDF of gaussian distribution by given fixed $\mu, \sigma$ parameters.

However, the second step to find the best step size $\rho_h$ is moderately demanding process. In the experiments, we tested 8 different choices for $p_h$ and selected the best one. Thus, it needs 8 forward passes of decoding and calculation of error between input and reconstruction.

## G  Additional Results

In this section, we present additional results of gain evaluated over different methods with Kodak and Clic-2021 dataset. Figure 9-12 shows th BD-rate gain for **bmshj2018-factorized** in Ballé et al. (2017), **mbt2018-mean** and **mbt2018** in Minnen et al. (2018) and **cheng2020-attn** in Cheng et al. (2020).

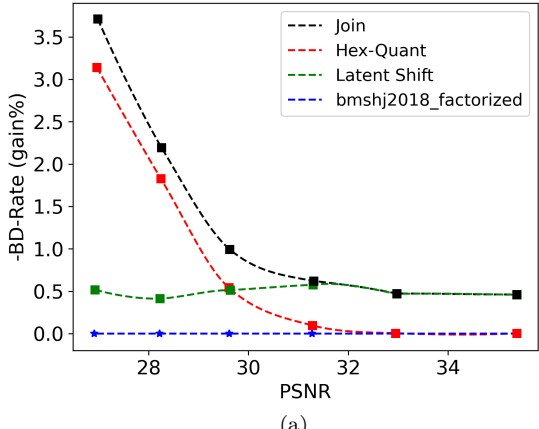
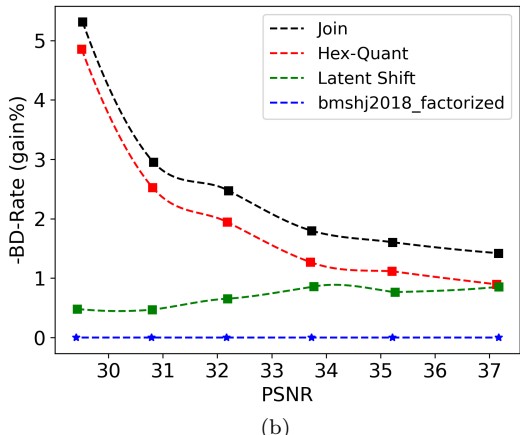

Figure 9: BD-Rates gain of our proposals from **bmshj2018-factorized** codecs for different quality a) Kodak test set b) Clic-2021

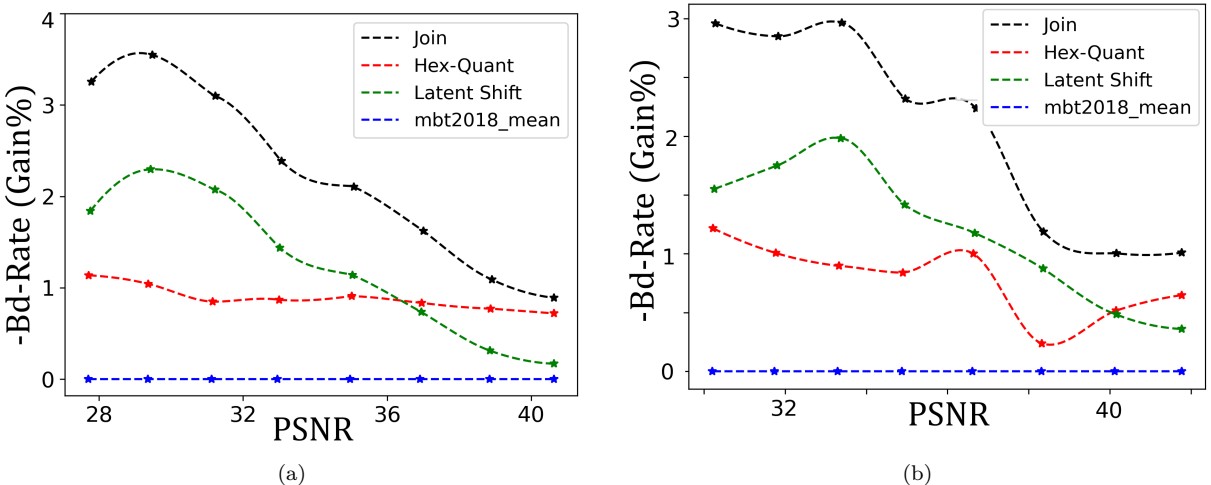

Figure 10: BD-Rates gain of our proposals from **mbt2018-mean** codecs for different quality a) Kodak test set b) Clic-2021

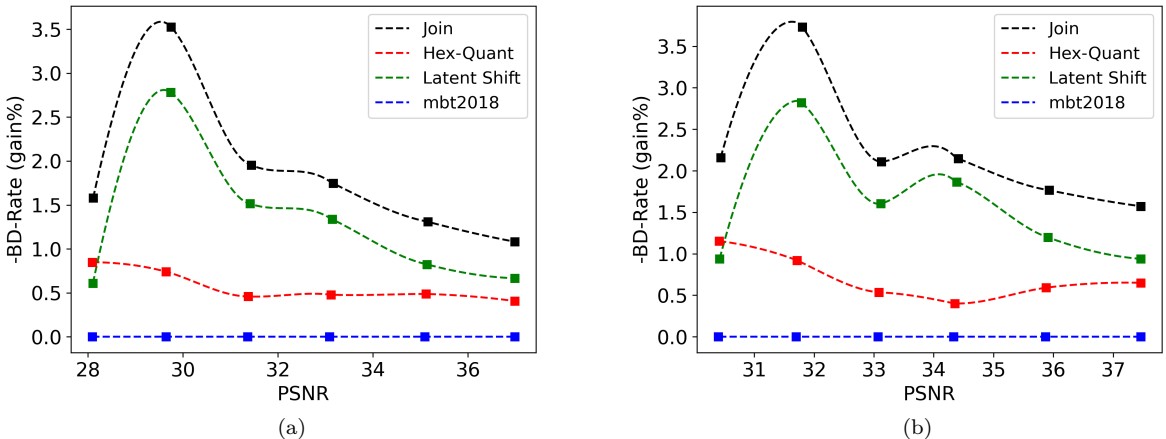

Figure 11: BD-Rates gain of our proposals from **mbt2018** codecs for different quality a) Kodak test set b) Clic-2021

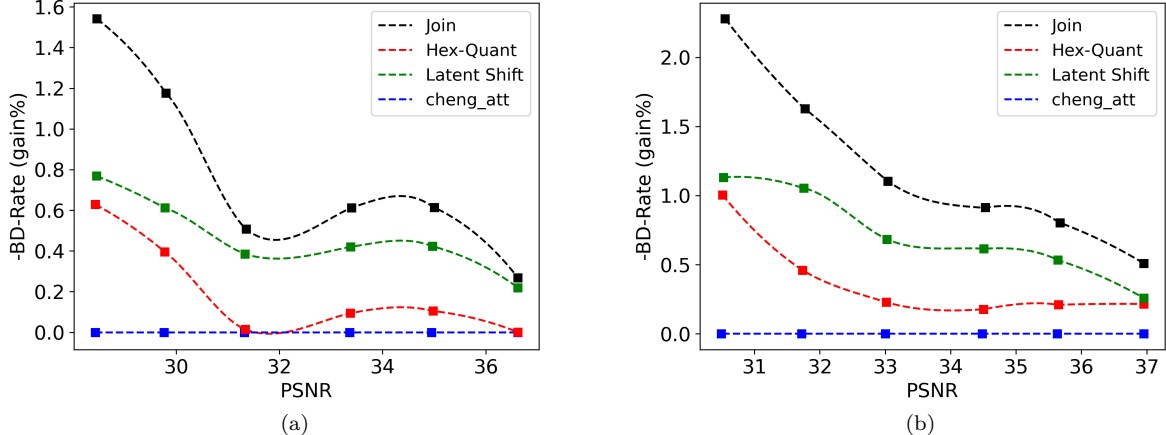

(a)  (b)

Figure 12: BD-Rates gain of our proposals from **cheng2020-attn** codecs for different quality a) Kodak test set b) Clic-2021

