# OpenReview forum: "Exploiting Latent Properties to Optimize Neural Codecs"
_TMLR — Rejected by TMLR_

### Review · Reviewer_jS5i · 2023-03-19

**Summary Of Contributions:**

The authors propose two improvements for neural image and video compression methods:
 - They use lattice instead of scalar quantization for the latents to reduce quantization error. Instead of learning a vector quantizer, the authors partition the latent vector into two and three-dimensional groups and apply the optimal uniform lattice quantizer.
 - They apply a linear shift to the decoded latent vector to reduce reconstruction error. Concretely, given a decoded latent vector $z$ and its probability $p(z \mid \theta)$ under the neural compression model, they use the score function $\nabla_z \log p(z \mid \theta)$ as the direction to shift $z$. According to the authors' proposal, the encoder conducts a line search along $\nabla_z \log p(z \mid \theta)$ to find the optimal shift, which they communicate to the decoder at minimal additional cost. The authors justify using the score as the direction by claiming that it correlates with the gradient of the distortion.

The authors demonstrate their proposals' utility by marginally improving the performance of popular off-the-shelf neural image and video compression models.

**Audience:**

Yes

**Broader Impact Concerns:**

No concerns.

**Claims And Evidence:**

No

**Requested Changes:**

 - [necessary] Could the authors clarify the relevance of Theorem 1 for the rest of the paper's contents?
 - [necessary] Could the authors clarify their claimed contribution in the uniform vector quantization sections? (Sections 3.1 and 3.2)
 - [necessary] Improve the writing; see the "Strengths and Weaknesses" section above.
 - [strongly recommended] Move some of the ablation studies from Appendix D to the main text.

**Strengths And Weaknesses:**

The authors' proposals could have good practical utility if we consistently save 1-4% of the bitrate for free, as claimed by the authors.

However, I worry that the amount of insight given in the paper is minimal and potentially flawed.

I don't see the relevance of Theorem 1. Its proof doesn't seem to give much insight into the behaviour of non-uniform quantization, and the authors don't appear to use it anywhere, either.

The authors propose to use optimal lattice quantizers to quantize the latent vector partitioned into two or three-dimensional groups. However, their proposal applies previously known results or straightforward extensions without seemingly providing any new insight.

Using the line search along the direction given by the score of the latents to improve the reconstruction error is an interesting idea. However, I am not convinced by the theoretical justification given for the correlation between the score and the gradient of the distortion. The expectation of their sum being $0$ doesn't imply they are correlated. For correlation, we require further information, e.g. about the expectation of cross-terms. Nonetheless, the authors demonstrate empirically that these terms are correlated. Appendix D compares line search along the score versus line search along a random direction, showing that the score direction is significantly better. The authors' setup seems somewhat questionable, as they use a standard Gaussian to sample directions; sampling from the unit n-dimensional sphere's surface makes more sense to me, but this is probably a minor detail. I found the random-versus-score direction comparison is the strongest point of the paper, and the authors should move this ablation study from the appendix to the main text.

Finally, I found the authors' experiments on the image and video datasets reasonable.

The writing needs improvement, as the paper is currently hard to read. There are many unusual phrases and strange sentence structures in the paper. Many of these errors could be fixed using a spell checker like Grammarly. To give a few examples:
 - "First, we prove that non-uniform quantization map on neural codec's latent is not necessary..."
 - "VQ learns non-uniform quantization center over the neural codec's latents which is doubly necessary..."
 - "Almost all complexity comes from finding the best shift step size by brutal force out of 8 candidates"

Some mathematical expressions contain errors, and some go against standard conventions, e.g.:
 - In section 3.2 the formula for $b$ is wrong, it should be $m^2 \cdot o$.
 - In Eq 10, $x$ and $y$ appear on the LHS as free variables, while on the RHS, they are bound by the integral. Furthermore, the authors use the star operator for pointwise multiplication, which is usually reserved for convolution.
 - Please use `\log` for logarithms instead of `log`.

Further minor issues:
 - The caption of Figure 1 needs improvement. Where is $f$? what are the $b$s and $c$s in the picture?
 - "Figure 2c shows that when the scale parameter of Gaussian increases, it approximates the theoretical known RD performance of uniform source" - it doesn't show the theoretical RD performance.
 - Figures 2a and 2b need more information. It is unclear how the authors obtained them.
 - Section 2: Why are we assuming square images?
 - In Section 2, paragraph starting "In this setting, the deep encoder..." - it would be good to have a picture describing this.

---

> ### Author Response · Authors · 2023-04-19
> **Relevance of Theorems and Novelty**
>
> We thank the reviewer for the insightful criticism of our paper.
> We list the arguments and our answers below.
>
>
>
> ## Misinterpreted Claims.
> We did not say anywhere that we save 1-4% of the bitrate for free. In exchange of computational complexity we have this gain. The computational costs are reported in Table 6. We are updating the main paper and put a section about computational complexity.
> But our savings are up to already trained model and we do not re-train the models. Thus, in this perspective our gains are free in terms of training time.
>
>
>
> ## Relevance of Theorem1.
> Theorem1 says if the encoder/decoder is enough expressive, we do not need non-uniform scalar quantization. Uniform scalar quantization is enough.  We indirectly use Theorem1 by assuming it is the same for vector quantization case. Thus, we use neither fixed non-uniform vector quantization map or learnable vector quantization map where all neural compression models with vector quantizer use. Thus, the one can see Theorem1 is the justification on why we select to use lattice (uniform) vector quantization map.
>
>
>
> ## Novelty on Lattice Quantization and its insights.
> We agree that Lattice Quantization is not new and studied very deeply in 90s. But that fact does not hurt the novelty of our proposal. Since in end-to-end compression literature, it was not used before. We will be glad if the reviewer can find a work that mentioned this point before us. Please also note that quantization is well studied between 1970-1990. Thus, all methods have very deep roots but still merit to use in new problems and techniques.  For instance, here are two recent works [1, 2] on end-to-end compression which uses Trellis Coded Quantization (TCQ) first introduced 1965 and used compression in 1990 and also Product Quantization has very long practical history but reformulated in 2010. The insight of using Vector Quantization is straight-forward. The lower quantization error the better reconstruction as we showed Appendix D3. Thus this paper opens new doors to decrease the quantization error.
>
>
>
> [1] SÜHRING, Karsten, SCHÄFER, Michael, PFAFF, Jonathan, et al. Trellis-Coded Quantization for End-to-End Learned Image Compression. In : 2022 IEEE International Conference on Image Processing (ICIP).
>
> [2] El-Nouby, A., Muckley, M. J., Ullrich, K., Laptev, I., Verbeek, J., & Jégou, H. (2022). Image Compression with Product Quantized Masked Image Modeling. TMLR, 2023
>
>
>
> ## Theoretical Justification of Correlation
> As we mentioned in the paper 2nd paragraph of page 7 clearly that KKT conditions cannot guaranty the correlation. We agree with the reviewer that for the guarantied correlation we may need further information. But at least this novel application of KKT conditions on neural codecs shows us why mentioned two gradients may have correlations. This theorem at least explains our experimentally discovered correlations between gradients.
>
>
>
> ## Ablation Studies on LatentShift
> We are totally on the same page with the reviewer on the necessity of the ablation studies. Indeed during preparing the paper, some of us wanted this ablation on main page as well since they clearly shows how the latent shift works. For the random direction, we tested many different random direction sampled from different distributions and put the best random direction result in the ablation study. Thus, this point is indeed minor. In addition to random versus gradient based shift, sign based shift is also very important in that ablation study. Because if there is no explicit gradient of the entropy but just general guess that "the higher the absolute value, the higher entropy" the one can use sign based shifting that we already tested it in traditional state-of-the art compression model (ECM 8.0) and saves 0.1% bd-rate which is very significant according to conventional standard research. We are updating paper with this result as well.
>
>
>
> ## Writings Improvement, Clarifying Figures
> We totally agree that we need to improve the writing, and all agree on mentioned minor notation errors.
> Assuming square image is for the simplicity and to reduce the number of different notions. There is no restriction about the image.
> Simple neural codec diagram would be great and we are working on it.
> Figure1 and 2 are going to be replaced with a new, clear figures and explanations.
> Especially on figure2, the known theoretical performance of 1,2,3 dimensional uniform VQ were given in Appendix B. Simply for dim=1, mse is 0.0833, dim=2, it is 0.0801 and dim=3, it is 0.0785 where MSE in the figure2c is approximating by increasing of the rate.

---

> > ### Comment · Reviewer_jS5i · 2023-04-29
> > **Response to the authors**
> >
> > I thank the authors for their rebuttal. I read the revised version of the paper, and while the authors address some of the issues I raised, I am unconvinced by all of the authors' responses. I address each outstanding point of the authors' responses below.
> >
> > ## Misinterpreted Claims
> > I realise that the way I wrote my statement was perhaps too vague, but thankfully the authors understood that I meant to say that their method can improve other methods "for free" without retraining.
> >
> > ## Relevance of Theorem 1
> > The justification "if it works in one dimension, it probably works in any dimension" in this case seems very questionable due to the problem's geometric nature. The geometry of one-dimensional spaces is well-known to differ significantly from the geometry of higher-dimensional spaces. In particular, higher-dimensional spaces lack a total order, which is a property the authors use in the proof of Thm 1. I am not saying that Thm 1 is uninteresting or insignificant or that exploring uniform VQ for compression is unsound. However, using Thm 1 to justify uniform VQ is invalid and potentially misleading. Thus, the authors should remove Thm 1 as a justification for their work.
> >
> > ## Theoretical Justification of Correlation
> > First and foremost, I should note that the fact that the empirical correlation found by the authors is valuable and interesting.
> >
> > However, the theoretical justification given by the authors is incorrect. The authors claim in their response that: "But at least this novel application of KKT conditions on neural codecs shows us why mentioned two gradients may have correlations." -- It does not. The fact that the expectation of a random variable is the negative of the expectation of another does not in any way imply that they are correlated. This claim must be removed for me to recommend publication.
> >
> > ## Writings Improvement, Clarifying Figures
> > While the writing has improved, I still find it difficult to read and needs more improvement. Furthermore, the caption of Figure 1 needs significant elaboration, as currently, the components in the figure have no explanation whatsoever.

---

> > > ### Author Response · Authors · 2023-05-01
> > > **Claims on the Theorems**
> > >
> > > We thank the Reviewer for the detailed explanation and valid objections on our claims. We are glad to hear that our correlation is valuable, the theorems are not at least uninteresting/insignificant and uniform VQ is not unsound.
> > >
> > > After getting Reviewer's valid criticism, we realized that our previous answer on Theorem2 is too vague, and it could not present our standing point well. In addition, we have some exaggerated words on our contribution in the paper. Since those are naturally counted as our claims by the Reviewer, to change those claims was demanded. We did some modification as listed below.
> > >
> > >
> > > ### Relevance of Theorem1
> > > Indeed, in our first submission, we presented the relevance of Theorem1 as how the Reviewer described: "if it works in one dimension, it probably works in any dimension". We agree that even though theorem is true, this relevance of the theorem was not true. Therefore, we changed our point on the relevance of Theorem1. We present the contribution of Theorem1 on eliminating all kind of SQ (all non-uniforms SQs) from the choice set. We have added a paragraph before Theorem1 that explains why non-uniform SQ is better than uniform SQ in general. But counter-intuitively, Theorem1 says it is pointless to try non-uniform SQ in neural codec. Thus, Theorem1 helps us to eliminate all SQ choices and proceed to VQ directly. We have also added two sentences at the end of the section3 to explain how Theorem1 helps us.
> > > We have also revised our contribution in Introduction and abstract accordingly.
> > > All changes are highlighted by blue font.
> > >
> > >
> > > ### Relevance of Theorem2
> > > As we mentioned before and also in the paper, Theorem2 never says anything about correlation. It just says the expectation of two gradients are zero. They show vice versa directions in expectations.  Thus, again Theorem2 is true.  But the relevance of the theorem was wrong because of two exaggerated words used to explain the contribution of this theorem.
> > >
> > > Here is how we see the relevance of Theorem2 to the paper. Theorem2 motivates us to check gradients' correlation. If you have two random variables that the expectations are zero, for sure it could not say anything about the correlation. But as an experimentalist, if the sum of two "random variable" (which we know that they are not independent) has zero expectation, the first idea comes into mind is to check their correlation. That is what we did and how we found the correlation. In order to clarify our point, we did some changes in the paper.
> > >
> > >
> > > Since KKT conditions shows some connection between two gradients but not correlation, we have modified a sentence in the introduction. Such that;
> > > "These conditions reveal a correlation ..." -> "These conditions reveal a connection..."
> > > Since correlation are experimentally explored after seeing this connection, we changed a sentence accordingly in the introduction.
> > > In section 4, we changed a word on why we used one gradient instead of another (not because of KKT conditions but because of experimentally found correlation). Again in section 4 , we put assumptions on how KKT can show correlation. These assumptions are; (i) if the training is on one single image (thus no expectation) and (ii) gradient's magnitude is not zero.
> > > all changes are highlighted by blue in the new version of the paper.
> > >
> > >
> > > ### Figure1 and Writing improvements
> > >
> > >
> > > We apologize for missing information in figure 1. We have improved the caption and added relatively enough information in the text with blue color.  We have improved the writings of our paper before and since there is a room for improvement, we are currently and constantly working on further enhancing the quality of the paper.

---

> > > > ### Comment · Reviewer_jS5i · 2023-05-03
> > > > **Response to the authors**
> > > >
> > > > I thank the authors for their response. I think the presentation of Theorem 1 is now fine, and the caption of Figure 1 is good.
> > > >
> > > > However, I checked the proof of Theorem 2 in Appendix C now, and I believe that the authors' claim is severely misleading compared to what they actually prove. Therefore, I cannot recommend the paper for publication.
> > > >
> > > > Concretely, as stated in the main text, Theorem 2 implies that the latents $\hat{z}$ and $\hat{y}$ depend on the input $x$, hence taking expectation over $x$ in Eq (3) and (4) makes sense. However, this is not what the authors prove in Appendix C. The authors prove that for fixed network weights, the optimal $z^*$ and $y^*$ which *do not* depend on $x$ satisfy
> > > > $$
> > > > \nabla_{\hat{z}}\log p_f(z^*) = -\nabla_{\hat{z}}\log p_h(y^* \mid z^*),
> > > > $$
> > > > and
> > > > $$
> > > > \nabla_{\hat{y}}\log p_h(y^* \mid z^*) = \lambda \mathbb{E}_{x \sim p_x}[ \nabla_\hat{y} d(x, g_s(y^*))].
> > > > $$
> > > > Unfortunately, these equations do not seem to give much actual insight and I think Thm 2 should be removed from the paper altogether unless the authors can correct their proof.

---

> > > > > ### Author Response · Authors · 2023-05-06
> > > > > **Error in proof of Th2**
> > > > >
> > > > > After carefully checking the proof and objection raised by the reviewer, indeed we realized that we mixed up proof on special case (KKT condition on encoder side finetuning solution, which is our follow-up work that we currently working on it) with general case. We truly appreciate for the attention on our proofs. We could not imagine more beneficial feedback than this one.
> > > > >
> > > > >
> > > > >
> > > > > Actually there is no need to fix the weights and no need to parameterize loss by instance specific main and side latent variables. It was the case in finetuning solution. That proof was not valid for either the general case or the fine-tuning case. Because there is also no expectation term in finetuning case (instance specific latents are optimized individually in finetuning case). We fixed proof by parameterize the loss function with network parameter and write KKT condition w.r.t network parameters. Later we multiply this KKT condition with Jacobian matrix (partial derivation of network parameter w.r.t side and main latent) and we reach the same conclusion.
> > > > >
> > > > > We slightly changed the Theorem2 and introduced a corollary belongs to Theorem2.  We updated the paper accordingly. Proof reading is still on going and we will upload proof read version soon.

---

### Review · Reviewer_MXQX · 2023-04-01

**Summary Of Contributions:**

In this paper, the authors propose two orthogonal methods to improve image/video compression with auto-encoders. To this end, the authors exploit two properties (i.e., vector quantization and entropy gradient-based latent shifting). Firstly, the authors use a uniform vector quantization map to improve the performance because they prove that the non-uniform quantization map is unnecessary. Secondly, they demonstrate that the gradient of entropy is correlated with the gradient of the reconstruction error. In the experiments, the proposed method has 2-4% rate savings at the same quality.

**Audience:**

Yes

**Claims And Evidence:**

Yes

**Requested Changes:**

1. The proposed method is able to improve the performances of many off-the-shelf codecs. In Table 1, the authors mainly compare older papers and one recent paper. It would be better to compare with more recent papers. In addition, recent diffusion models are very popular and they also use convolution or Transformer-based auto-encoders. Can the proposed method be used in diffusion models? Could you provide some discussions about this?

2. The proposed method improves the performance using the uniform VQ and the Latent Shift. However, they will also introduce additional complexity during encoding and decoding over the baseline methods. Could you provide some discussions on how to improve this issue?

**Strengths And Weaknesses:**

Strengths:
1. This paper provides theoretical insights into the latent properties.
2. The proposed method can be used in many image/video compression networks to improve performance.
3. The proposed method saves 2-4% of the bit-rate for various pre-trained methods.

Weaknesses:
1. The proposed method introduces additional complexity during encoding and decoding.

---

> ### Author Response · Authors · 2023-04-19
> **New Comparisons and Complexity**
>
> We thank our reviewer for the comments, and we are glad that the reviewer appreciated our method as novel and theoretically sound. Below we address the concerns of the reviewer.
>
>
>
> ## New Comparisons
> Since the recent neural compression methods goes in the direction of increasing the complexity to achieve better rate-distortion trade-off, we could not find comparisons with more recent work as useful. One of the main drawbacks of neural compression to be not practically available is its decoder complexity. Even the simplest neural compression models' decoder is x500-x1000 more complex than conventional compression standard such as VVC or new coming standard ECM. Indeed, the model that we added as recent work for instance is x5-10 more complex than well-known neural baselines. That is why we respectfully think that this recent model and 4-5 strong well-known baseline models are enough to test. Further, as mentioned in the paper, our proposal is generic and could be applied on any pre-trained neural codec.
>
> However, after the submission, we explored that our LatentShift can also improve conventional compression model as well. We tested it up to the current state of the art ECM.8.0 and find 0.1% bd-gain without changing encoder and decoder complexity. This 0.1% bd-gain is very significant according to the standard community and hopefully it will be adapted to the standard. In this way, our LatentShift will be the very first model that traditional compression model borrows from the neural compression models. So far, the transition was vice-versa. Neural compression models borrowed many techniques from conventional compression.
>
>
>
> ## LatentShift on Diffusion Models
> Recent diffusion models uses "score function" which is exactly gradients of rate (entropy). However, using KKT conditions, we showed that this gradient (we do not directly care about it) is somehow correlated with gradient wrt distortion which we really care. Thus, we use one as it is negative of another. We can get this result because in loss function, there are two objectives. Thus, one should be negative of another. But how to improve generative performance of diffusion models by KKT conditions is out of our knowledge.
>
> In addition, diffusion model is also used in compression tasks as well. In compression task, diffusion model is used as the decoder to reconstruct images as far as we are aware of, thus our latent shift is applicable on the diffusion models also.
>
>
>
> ## Complexity
> We gave some complexity analysis in AppendixF2. We are updating the paper and put it into main text with some additional comments and tests. Thanks to the fast quantization on high dimensional vectoral grids, our quantization gives negligible extra complexity. But the main complexity source is to find the best step size in encoding time for LatentShift. Since neural codecs encoder complexity is way smaller than conventional codecs, and in many applications, encoding complexity is not the issue, we may neglect this extra encoding complexity. Again, in decoding time, Latent shift just needs to calculate gradient of entropy and in most of the model, there is closed form solution of gradient of the entropy since the probability model is one of the known parametric models such as Gaussian or Laplacian.
>
> On the other hand, mentioned encoder complexity can be even removed by using one single universal step size. Thus, we do not need to do step size search in encoding time. But it could also decrease the gain that we potentially get. We implemented Latentshift on conventional compression model with universal step size.

---

### Review · Reviewer_BrCz · 2023-04-05

**Summary Of Contributions:**

The paper proposes two procedures that can be used with any pre-trained compressive auto-encoder to improve its performance. Both of these procedures are supported by proofs, empirical analysis, and experiments.

The first contribution is the finding using uniform vector quantization can improve compression performance over methods using uniform rounding or non-uniform vector quantization.

The second contribution is the finding that the gradient of the rate loss is correlated with the gradient of the distortion loss. This is proved to be true on average and empirically shown to be true for individual data points as well. With a little encoder-side optimization and a negligible rate overhead, this information can be exploited for better R/D performance.

**Audience:**

Yes

**Claims And Evidence:**

Yes

**Requested Changes:**

## Writing style

- Please pay attention to using articles in sentences. In the abstract alone many are missing:
    - “state of the art neural codecs do not take advantage of **the/a** vector quantization technique”
    - “we prove that **a** non-uniform quantization map on **the** neural codec’s latent is not necessary”
    - “we theoretically show that **the** gradient of **the** entropy”
- Many sentences could be formulated more precisely, e.g.

    >  “End-to-end deep compression methods are generally rate-distortion autoencoders (Habibian et al., 2019), where the latents are optimized”

    -    the latents are not optimized, the model (consisting of encoder, hyperrior, prior, and decoder) is optimized.

    > “, the hyperprior entropy p_h”
    -  “, the hyperprior entropy **model** p_h”

    > “VQ learns non-uniform quantization center over the neural codec’s latents which is **doubly necessary**”
    - I don't know what is meant by this sentence.

    >  “the length of the bitstream  $\tilde{y}$.”
    - “The length of the bitstream *when entropy coding* $\tilde{y}$.” since $\tilde{y}$ is not the bitstream, it is a tensor.


   > “all current methods implement quantization as a 1-bin width uniform Scalar Quantization (SQ).”

    - Indeed most do, but not all. For example:
        - Mentzer et al. (2018) and Habibian et al. (2019) use learned scalar quantization.
        - Agustsson et al. (2017) use learned vector quantization.

  >  “they are only almost on par with the best traditional image compressing methods when trained to optimized the peak signal-to-noise ratio (PSNR) according to our knowledge.”

    - Specifically for image compression there are methods that outperform VVC (e.g. He et al., 2022).
    - "trained to ~~optimized~~ **optimize**"
    - While _evaluated_ on PSNR, models are typically trained to _optimize_ MSE.

 - In Table 2, please show the average for UVG under all the UVG videos instead of in the middle.

All in all I recommend the authors to do some more passes over the text and ask for help proofreading the paper.

## Questions
- In Theorem 1. could you specify more clearly which probability distribution you are referring to (e.g. with a subscript)? Am I correct in interpreting the proof as follows?

  $$\int_{i-0.5}^{i+0.5} p_Z(z)d(z) = \int_{f^{-1}(i - 0.5)}^{f^{-1}(i + 0.5)} p_Z(f(y))f'(y)dy =  \int_{f^{-1}(i - 0.5)}^{f^{-1}(i + 0.5)} p_Y(y)dy $$

- For the uniform vector quantization it seems like the quantization procedure is correlating the latent dimensions (as can be seen for example in Figure 2c).
    - The experiments are performed for 1 (SQ), 2 (hexagon), and 3 (truncated octahedron) dimensions. Could this be generalized to more dimensions and would the authors expect larger improvements?
    - Since the model is not trained end-to-end with the quantization method it is an open question of which dimensions to group together (into the dimension named “v” in the paper). The authors seem to use a heuristic based on the value of sigma. Do the authors have intuitions about the importance of choosing which dimensions to group together?


### Citations
- Mentzer, Fabian, et al. "Conditional probability models for deep image compression." Proceedings of the IEEE Conference on Computer Vision and Pattern Recognition. 2018.
- Habibian, Amirhossein, et al. "Video compression with rate-distortion autoencoders." Proceedings of the IEEE/CVF International Conference on Computer Vision. 2019.
- Agustsson, Eirikur, et al. "Soft-to-hard vector quantization for end-to-end learning compressible representations." Advances in neural information processing systems 30 (2017).
- He, Dailan, et al. "Elic: Efficient learned image compression with unevenly grouped space-channel contextual adaptive coding." Proceedings of the IEEE/CVF Conference on Computer Vision and Pattern Recognition. 2022.


**Strengths And Weaknesses:**

## Strengths
- Both techniques are creative and novel contributions.
- As the authors demonstrate, these techniques can be applied to most of the standard compression models without too much overhead, and can immediately improve compression performance.
- The paper contains elaborate proofs, analysis, and experiments to support the conclusions, including run-time analysis.
- Experiments are conducted on several state-of-the-art image and video compression models, demonstrating their effectiveness in a real-life setting.


## Weakness
- Especially for the latent shift, the performance improvement is not very big. That said, the interventions are relatively minor.
- Relation to existing methods is sometimes a bit underexplored:
    - Gradient of Entropy: Since some encoder-side optimization is required here the question arises of how orthogonal this approach is to direct latent optimization (Campos et al., 2019).
    - Uniform vector quantization: Although it is nice that the authors show that this method can be used for models trained with uniform scalar quantization, little to no reflection is given on training with uniform vector quantization. This is especially relevant since a large portion of the compression field is dedicated to how to best train with quantization proxys (Agustsson et al., 2020; Yang et al., 2022; Guo et al.).
- The writing style of the paper can be improved. See the next section for details.

### Citations
- Joaquim Campos, Simon Meierhans, Abdelaziz Djelouah, and Christopher Schroers. Content adaptive optimization for neural image compression. In Proceedings of the IEEE/CVF Conference on Computer Vision and Pattern Recognition (CVPR) Workshops, pp. 0–0, 2019.
- Yang, Yibo, Robert Bamler, and Stephan Mandt. "Improving inference for neural image compression." Advances in Neural Information Processing Systems 33 (2020): 573-584.
- Agustsson, Eirikur, and Lucas Theis. "Universally quantized neural compression." Advances in neural information processing systems 33 (2020): 12367-12376.
- Guo, Zongyu, et al. "Soft then hard: Rethinking the quantization in neural image compression." International Conference on Machine Learning. PMLR, 2021.

---

> ### Author Response · Authors · 2023-04-20
> **Connection to the Finetuning Solutions**
>
> We cannot enough thank to the reviewer's insightful questions and glad to hear that our proposals are found creative and novel.
> Below we address the issues that raised by our reviewer.
>
> ## Performance of LatentShift
> Even if the performance is not that big according to general machine learning researches, it is very significant in compression researches.  We tested it on traditional codecs (on top of ECM8.0) and it improved the bd-rate by 0.1% where standard community count it significant increment and hopefully it will be adapted to the standard. LatentShift will be one of the very first approaches that traditional method borrow from neural codecs. We are updating the paper according to this new results.
>
>
> ## Connection to the Finetuning Solutions
> There is strong connection to encoder side image specific optimization (finetuning) solutions such as (Campos et al., 2019). Actually we studied the connection deeply but we let it as future work on Latent Shift.
>
> First point is that encoders side optimization solutions increases the encoder complexity enormously. It is reported that they are x400 times more complex even with massive parallelization (Yang 2020, Campos 2019).  Since we think the single thread run without parallelization is a good proxy of energy consumption, we implemented these solution and figure out that they need around x4000 times more encoding time when single thread is considered. Since our solution is just search the best step size out of 8 candidates, it needs x8-x10 more encoding time on single thread which can be decrease to x2 by parallelization.
>
> Second and the most important point is the orthogonality of LatentShift to the Finetuning solutions. Even though finetuning solutions has more gain compare to LatentShift, it cannot compensate our gain. In addition, after finetuning solution, LatentShift even saves more bits than only LatentShift.
>
> The reason is that finetuning solution optimizes the latents for the same loss Eq(1) but for single image. Thus in finetuning solutions' loss  there is no expectation terms over training set. So the KKT conditions (gradients show vice -versa directions) has no expectation terms over training set as well in Eq(3-4) which makes their correlations closer to -1.
>
> For our future work, we tested LatentShift only, finetuning only (Campos 2019) and  finetuning and Latentshift together. Here are some initial results
>
> | Model | Encoding Time | Decoding Time | Bd-Rate (Kodak) | Correlation (Kodak) | Bd-Rate (Clic)| Correlation (Clic) |
> | --- | :----:  | :----:  | :---: | :---: |:---:| ---- |
> | mbt2018-mean (baseline)  | 0.5274sec| 0.7138sec | 0% | N/A | 0% | N/A |
> | mbt2018-mean + LatentShift | 5.5674sec (x10.1)| 0.7188sec (+0.7%) | -1.27% | -0.1805 | -1.21% | -0.1465|
> | mbt2018-mean + FineTuning|    2312sec (x4380)  | 0.7138sec (+0.0%) | -5.77% | N/A  | -5.53% | N/A |
> | mbt2018-mean + FineTuning + LatentShift  |  2317 (4390x)  | 0.7188sec (+0.7%) | -7.27% | -0.2012 | -6.96% | -0.1596|
>
> It can be seen that LatentShift can saves even more bits after finetuning solutions.
>
> ## Train model with Vector Quantization
> Even though we decrease average quantization error with VQ, for some instance, the quantization error is increased and it goes beyond the trained noise range for quantization. ( for SQ it is uniform noise from -0.5 to 0.5) So the trained decoder never saw that individual high error during training.  As our reviewer says the best practice would be define a proper relaxation and train the model which will be our future work.
>
> ## Writing Styles and Citations
> We all agree on the recommendation for writing, notations..etc. They are going to be fixed in the new version.
>
> ## Going beyond 3 Dimension of VQ
> In theory, we can reach theoretical RD limit by infinite number of dimensional VQ. Theoretically, we can monotonically improve the performance by increasing the dimension. In practice maximum 24-dimensional Lattice (Leech lattice) is used. Because of the simplicity, we showed improvement for 2 and 3 dimension. It is straight forward to extend the dimension. In exchange of encoder decoder complexity, we expect more gain. This point is also in our future work.
>
> ## How to group latent into vector
> In theory, performance of quantization should not be effected by selection of grouping. But we select to grouping them by their distribution is because limiting the necessity of PMF table to be kept, not because of quantization perspective. For 2dim case, if two latent comes from any distributions, we need 2-dim PMF tables for all possible pair of sigma. Since we group the latents from the same distribution, the number of PMF tables are the same in default model.
>
> ## Which Probability Distribution
> Thanks for the interpretation. That is totally correct. Indeed it is way more beautiful than what we used. We do not have any restriction on  $p_Y(.)$ and it can be any pdf learned by default model. But as you showed,  $p_Z(.)$ depends on $p_Y(.)$ and the transform.

---

> > ### Comment · Reviewer_BrCz · 2023-04-25
> > **Thanks for the response. It would be helpful to see a second revision.**
> >
> > I would like to thank the authors for their elaborate and detailed responses. The latent finetuning experiment is a great addition to the paper and makes the case for LatentShift even stronger.
> >
> > Taken all together I think the conceptual and experimental contribution of the paper is enough to warrant publication and only the concerns about the writing quality remain. Given that other reviewers also raised these concerns, it would be very helpful if the authors could upload a new revision of their manuscript before the official recommendation deadline.
> >
> > Some more minor points:
> > - I encourage the authors to reflect on the applicability of their proof as suggested by reviewer MXQX.
> > - In the experiment section, the authors state that BD-rate is calculated over a certain bitrate range. However for a fair comparison, the distortion interval should be fixed, not the rate interval so that we compare rate savings over the same distortion range. Can the authors specify more clearly how they computed the BD scores?

---

> > > ### Author Response · Authors · 2023-04-28
> > > **bd-rate calculations**
> > >
> > > Thanks for pointing out the bd-rate calculations.
> > >
> > > That is truly correct point that you take our attention. Actually we calculate bd-rate on overlapped PSNR range for each image as you mentioned. Honestly we do not have any control of bd-rate calculations. According to standard researches/contributions, performance metrics are strictly determined under Common Test Conditions (CTC) [1]. We can feed our psnr and our bitlength along with baseline psnr and bitlength to the metric module. What we meant in the text that we used that certain rate range is for giving an idea about tested model. This given rate range is the average rates from the model that gives most quality reconstruction to the model that gives least quality reconstruction. So we do not have any control of these model since we did not train the model. These range is the provided models range in average.
> > >
> > > We already uploaded a new version and fixed this point along with another demands in the main text.
> > >
> > > [1] Marta Karczewicz and Yan Ye. Common test conditions and evaluation procedures for enhanced compression tool testing. JVET-Y2017, 2022.

---

### Author Response · Authors · 2023-04-28
**Revised Paper uploaded**

We would like to thank to all the reviewers for their time and attention to our paper.
Indeed the feedbacks are very helpful.
We answered almost all of the demands and revised the paper accordingly.

---

### Author Response · Authors · 2024-11-07
**IEEE publication policy**

This work has been submitted to the IEEE for possible publication. Copyright may be transferred without notice, after which this version may no longer be accessible.

---

### Decision · Action_Editors · 2023-06-02

**Recommendation:** Reject

**Comment:**

First of all, I would like to thank the reviewers for their effort to improve the paper's quality.
I carefully read all reviews and also recommendations. All reviewers are in agreement that there are multiple points that should be improved. The three of them lean toward rejection. Their reasoning could be summarized in the following way:
- The writing, experiment section, and theoretical justification should be improved.
- The experimental evidence is interesting to the audience, but the results are not properly presented.
- The paper presents some interesting empirical findings, but some of the theoretical analysis is extremely flawed. This is unfortunately exacerbated by quite poor writing quality.

As a result, I regret to inform you that it is a clear case in which I cannot recommend accepting the paper.

**Audience:**

The paper discusses a relevant topic to the TMLR's audience.

**Claims And Evidence:**

The paper has two claims:
1. The scalar quantization could be replaced with the uniform vector quantization.
2.  The KKT conditions are applied to a neural codec.

However, the reviewers raised some issues regarding evidence:
- Especially for the latent shift, the performance improvement is not very big.
- The proposed method introduces additional complexity during encoding and decoding.
- The proof of Theorem 2 is invalid.

All three reviewers are in agreement that the paper should be improved and is not ready for publishing.